# Temporal Contrastive Learning through implicit non-equilibrium memory

Martin J. Falk [1,3], Adam T. Strupp[1,3], Benjamin Scellier[2] & Arvind Murugan [1] ✉

The backpropagation method has enabled transformative uses of neural networks. Alternatively, for energy-based models, local learning methods involving only nearby neurons offer benefits in terms of decentralized training, and allow for the possibility of learning in computationally-constrained substrates. One class of local learning methods *contrasts* the desired, clamped behavior with spontaneous, free behavior. However, directly contrasting free and clamped behaviors requires explicit memory. Here, we introduce 'Temporal Contrastive Learning', an approach that uses integral feedback in each learning degree of freedom to provide a simple form of implicit non-equilibrium memory. During training, free and clamped behaviors are shown in a sawtooth-like protocol over time. When combined with integral feedback dynamics, these alternating temporal protocols generate an implicit memory necessary for comparing free and clamped behaviors, broadening the range of physical and biological systems capable of contrastive learning. Finally, we show that non-equilibrium dissipation improves learning quality and determine a Landauer-like energy cost of contrastive learning through physical dynamics.

The modern success of neural networks is underpinned by the backpropagation algorithm, which easily computes gradients of cost functions on GPUs[1]. Backpropagation is a 'non-local' operation, requiring a central processor to coordinate changes to a synapse that can depend on the state of neurons far away from the synapse. While powerful, such methods may not be available in physical or biological systems with strong constraints on computation and communication.

A distinct thread of learning theory has sought 'local' learning rules, such as the Hebbian rule ('fire together, wire together'), where updates to a synapse are based only on the state of adjacent neurons (Fig. 1). Such local rules allow for, e.g. distributed neuromorphic computation[2–4]. Excitingly, local learning rules also open the possibility of endowing computationally-constrained physical systems with functionality through an in situ period of training, rather than by prior backpropagation-aided design on a computer[5–21]. In these settings, sometimes as simple as chemical reactions within a cell or a mechanical material, there is no centralized control that would allow backpropagation to be a viable method of learning. Consequently, local rules allow for the intriguing possibility of autonomous 'physical

learning'[5,14] — no computers or electronics needed — in a range of both natural and artificial systems of constrained complexity such as molecular and mechanical networks.

In particular, a large class of local 'contrastive learning' algorithms (contrastive Hebbian learning[22–24], Contrastive Divergence[25], Equilibrium Propagation[26]) promise impressive results, but make requirements on the capabilities of a single synapse (or more generally, on learning degrees of freedom). While details differ, training weights generally are updated based on the difference between Hebbian-like rules applied during a 'clamped' state that roughly corresponds to desired behaviors and a 'free' state that corresponds to the spontaneous (and initially undesirable) behaviors of the system.

Therefore, a central obstacle for autonomous physical systems to exploit contrastive learning is that weight updates require a comparison between free and clamped states, but these states occur at different moments in time. Such a comparison requires memory at each synapse to store free and clamped state information in addition to global signals that switch between these free and clamped memory units and then retrieve information from them to perform weight

[1]Department of Physics, University of Chicago, Chicago, IL, USA. [2]Rain AI, San Francisco, IL, USA. [3]These authors contributed equally: Martin J. Falk, Adam T. Strupp. ✉e-mail: amurugan@uchicago.edu

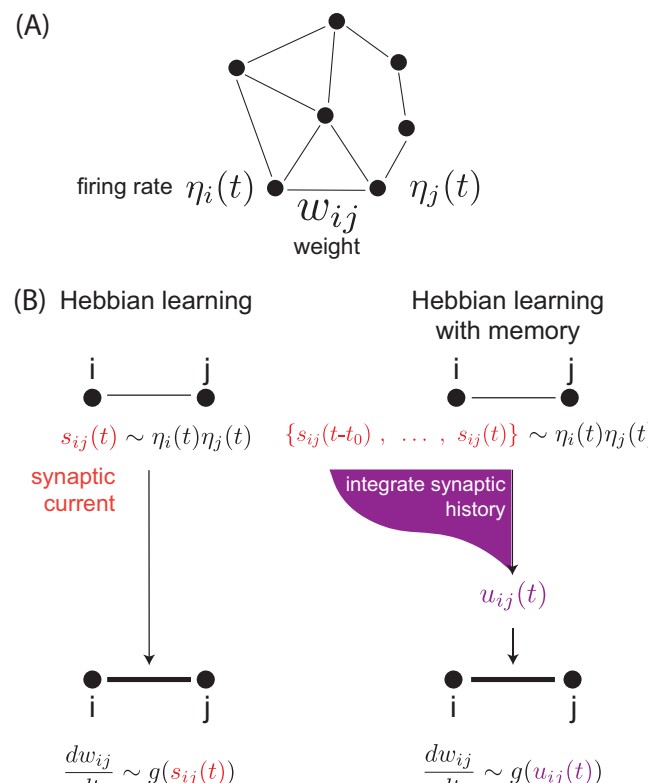

**Fig. 1 | Hebbian learning with memory. A** Consider a neural network with neuron $i$ firing at rate $\eta_i(t)$ and synaptic weight $w_{ij}$ between neurons $i, j$. **B** In Hebbian learning, the weights $w_{ij}$ are changed as a function $g$ of the synaptic current $s_{ij}(t) \cdot \eta_i(t)\eta_j(t)$. Here, we generalize the Hebbian framework to a model in which weights $w_{ij}$ are changed based on the history of $s_{ij}(t)$, i.e., based on $u_{ij}(t) = \int_{-\infty}^{t} K(t - t')s_{ij}(t')dt'$ where $K$ is a memory kernel. We find that non-monotonic kernels $K$ that arise in non-equilibrium systems encode memory that naturally enables contrastive learning through local rules.

updates[27,28]. These requirements make it difficult to see how contrastive learning can arise in natural physical and biological systems and demand additional complexity in engineered neuromorphic platforms.

Here, our primary contribution is to show how a ubiquitous process - integral feedback control - can allow for contrastive learning without the complexities of explicit memory of free and clamped states or switching between Hebbian and anti-Hebbian update modes. Further, in our method, computing the contrastive weight update signal and then performing the update of weights are not separate steps involving different kinds of hardware but are the same unified in situ operation. This approach to contrastive learning, which we call 'Temporal Contrastive Learning through feedback control', allows a wide range of physical and biological platforms without central processors to physically learn (i.e., autonomously learn) novel functions through contrastive learning methods.

First, we introduce a simple model of implicit memory in integral feedback-based update dynamics at synapses. Synaptic weights $w_{ij}$ are effectively updated by $\int_{-\infty}^{t} K(t - t')s_{ij}(t')dt'$, where $K$ is a memory kernel. Updates of this form arise naturally due to integral feedback control[29] of synaptic current $s_{ij}$, without the need for an explicit memory element.

Using a microscopically reversible model of the dissipative dynamics at each synapse, we are able to calculate an energy dissipation cost of contrastive learning. The dissipation can be interpreted as the cost of (implicitly) storing and erasing information about free and clamped states over repeated cycles of learning.

Finally, we propose how the non-equilibrium memory needed for contrastive learning arises naturally as a by-product of integral feedback control[29–31] in many different physical systems. As a consequence, a wide range of physical and biological systems might have a latent ability to learn through contrastive schemes by exploiting feedback dynamics.

## Background: Contrastive Learning

Contrastive Learning (CL) was introduced in the context of training Boltzmann machines[32] and has been developed further in numerous works[22–26].

While CL was originally introduced and subsequently developed for stochastic systems (such as Boltzmann machines)[25,32,33], here we review the version of [22,26] for deterministic systems (such as Hopfield networks). CL applies in systems described by an energy function $E$ (more accurately, a Lyapunov function). The system may be supplied with a boundary input and evolves towards a minimum of $E$, called 'free state'. The system may also be supplied with a boundary desired output (in addition to the boundary input), driving the system towards a new energy minimum, called 'clamped state'. Contrastive training requires updating weights as:

$$\Delta w_{ij} = \epsilon \left( s_{ij}^{\text{clamped}} - s_{ij}^{\text{free}} \right), \tag{1}$$

where $\epsilon$ is a learning rate and $s_{ij}^{\text{free}}, s_{ij}^{\text{clamped}}$ are the synaptic currents at the free and clamped states, defined by $s_{ij} = -\frac{\partial E}{\partial w_{ij}}$. For example, for a neural network with a Hopfield-like energy function, we recover a conventional Hebbian rule with $s_{ij} = \eta_i \eta_j$, where $\eta_i$ is the firing rate of neuron $i$ (Fig. 1B).

A key benefit of contrastive training is that it is a *local* rule; synaptic weight $w_{ij}$ is only updated based on the state of neighboring neurons $i, j$. Hence, it has been proposed as biologically plausible[34] and also plausible in physical systems[9,14]. However, a learning rule of this kind that compares two different states raises some challenges. Experiencing the two states and updating weights in sequence, i.e., $\Delta w_{ij} = -\epsilon s_{ij}^{\text{free}}$, followed by $\Delta w_{ij} = \epsilon s_{ij}^{\text{clamped}}$, will require a very small learning rate $\epsilon$ to avoid convergence problems since the difference $s_{ij}^{\text{clamped}} - s_{ij}^{\text{free}}$ can be much smaller than either of the two terms[35]. Further, this approach requires globally switching the system between Hebbian and anti-Hebbian learning rules. One natural option then is to store information about the free state synaptic currents $s_{ij}^{\text{free}}$ locally at each synapse during the free state - without making any weight updates - and then store information about the clamped state currents $s_{ij}^{\text{clamped}}$, again without making any weight updates, and then after a cycle of such states, update weights based on Eq. (1) using the stored information. While this approach is natural when implementing such training on a computer, it imposes multiple requirements on the physical or biological system: (1) local memory units at each synapse that stores information about $s_{ij}^{\text{free}}, s_{ij}^{\text{clamped}}$ before $w_{ij}$ are updated, (2) a global signal informing the system whether the current state corresponds to free or clamped state (since the corresponding $s_{ij}$ must be stored with signs, given the form of Eq. (1)). These requirements can limit the relevance of contrastive training to natural physical and biological systems, even if they can naturally update parameters through Hebbian-like rules[5,8,11,13,23].

## Results

### Temporal Contrastive learning using implicit memory

We investigate an alternative model of implicit non-equilibrium memory, Temporal Contrastive Learning (TCL) in a synapse (or any equivalent physical learning degree of freedom[5]). In our model, clamped or free states information is not stored anywhere explicitly; instead, non-equilibrium dynamics at each synapse results in changes to $w_{ij}$ that are based on the difference between past and present states.

Consider a neural network driven to experiencing a temporal sequence of inputs (e.g., smoothly interpolating between free and

clamped states according to a periodic temporal protocol) which induces a synaptic current $s_{ij}(t)$ at synapse $ij$. For example, the Hebbian rule ('fire together, wire together') is based on a signal $s_{ij}(t) = \eta_i(t)\eta_j(t)$ where $\eta_i$ is the firing rate of neuron $i$[36,37]. The conventional Hebbian rule assumes an instantaneous update of weights with the current value of the synaptic signal,

$$\frac{dw_{ij}}{dt} = g(s_{ij}(t)), \qquad (2)$$

where $g$ is a system-dependent non-linear function[36].

In contrast, we consider updates based on an implicit memory of recent history $s_{ij}(t)$, with the memory encoded by convolution with a kernel $K(t - t')$. This kernel convolution arises through the underlying physical dynamics in diverse physical and biological systems to be discussed later; in this approach, there is no explicit storage of the value of the signal $s_{ij}(t)$ at each point in time in specialized memory.

The memory kernel, together with the past signal, characterizes the response of each synapse to the history of signal values:

$$u_{ij}(t) = \int_{-\infty}^{t} K(t - t') s_{ij}(t') \, dt' \qquad (3)$$

$$\frac{dw_{ij}}{dt} = g(u_{ij}(t)), \qquad (4)$$

where the nonlinearity $g$ may be system-dependent.

While most physical and biological systems have some form of memory, the most common memory is described by monotonic kernels, for example $K(t - t') \sim e^{-(t-t')/\tau_K}$. But a broad class of non-equilibrium systems exhibit memory with a non-monotonic kernel with both positive and negative lobes, for example $K(t - t') \sim e^{-(t-t')/\tau_K} f(t - t')$, with $f(t - t')$ a polynomial[31,38–45]. Such kernels arise naturally in numerous physical systems, the prototypical example being integral feedback[29] in analog circuits.

A key intuition for considering such an update rule can be seen in the response of the system to a step-function signal; when convolved with a non-monotonic kernel, the constant parts of the signal produce a vanishing output. However, at the transition from one signal value to another, the convolution produces a jump with a timescale inherent to the kernel (Fig. 2B). As such, the convolution in effect produces the finite-time derivative of the original signal.

Finally, $g(u)$ in Eq. (4) can represent any non-linearity that suppresses small inputs $u$ relative to larger $u$, e.g.,

$$g(u) = \begin{cases} u & |u| \geq \theta_g \\ 0 & |u| < \theta_g \end{cases}. \qquad (5)$$

This non-linearity allows for differentiation between fast and slow changes in synaptic signal.

The weight update of the system described thus far does not change in time, e.g., between clamped and free states; synapses are always updated with the same fixed local learning rule in Eq. (4).

The only time-dependence comes from training examples being presented in a time-dependent way. For simplicity, we consider a sawtooth-like training protocol, where input and output neurons $\eta_i(t)$ alternate between free and clamped states over time (Fig. 2C). In this sawtooth protocol, the free-to-clamped change is fast (time $\tau_f$) while the clamped-to-free relaxation is slow (time $\tau_s > \tau_f$); see Section S1 for further detail.

Such a time-dependent sawtooth presentation of training examples $\eta_i(t)$ induces a sawtooth synaptic signal $s_{ij}(t)$. We can compute the resulting change in weights $\Delta w_{ij}$, which is Eq. (4) integrated over one cycle of the sawtooth, assuming slow weight updates:

$$\Delta w_{ij} = \epsilon \int_0^{\tau_f + \tau_s} g\left( \int_{-\infty}^{t} K(t - t') s_{ij}(t') \, dt' \right) dt, \qquad (6)$$

with $\epsilon$ a learning rate. As shown in Section S1, in the specific limit of protocols with $\tau_K \ll \tau_f \ll \tau_s$:

$$\Delta w_{ij} \approx \epsilon \left( s_{ij}^{\text{clamped}} - s_{ij}^{\text{free}} \right). \qquad (7)$$

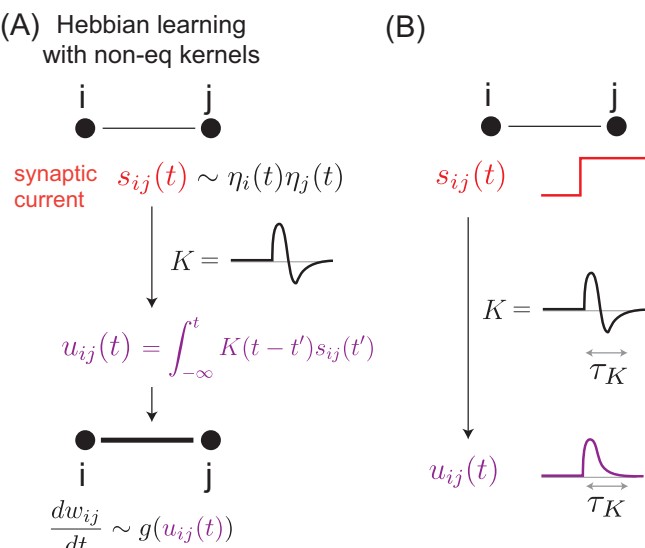

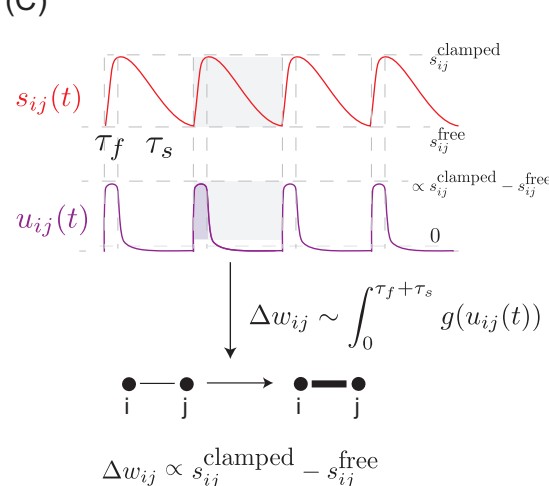

**Fig. 2 | Contrastive learning through non-monotonic memory kernels. A** We consider a network whose weights $w_{ij}$ are updated by a history of the synaptic current $s_{ij}(t)$, i.e., by $u_{ij}(t) = \int_{-\infty}^{t} K(t - t') s_{ij}(t') dt'$ with a non-monotonic kernel $K$ of timescale $\tau_K$ as shown. $g(u)$ is a non-linear function of the type in Eq. (5). **B** Response of $u_{ij}$ to a step change in $s_{ij}$ shows that $u_{ij}$ effectively computes a finite-timescale derivative of $s_{ij}(t)$; $u_{ij}$ responds to fast changes in $s_{ij}$ but is insensitive to slow or constant values of $s_{ij}(t)$. **C** Training protocol: Free and clamped states are seen in sequence, with the free state rapidly followed by the clamped state (fast timescale $\tau_f$) and clamped boundary conditions slowly relaxing back to free (slow timescale $\tau_s$). The rapid rise results in a large $u_{ij}(t)$ whose integral reflects the desired contrastive signal; but the slow fall results in a small $u_{ij}(t)$ whose impact on $w_{ij}$ is negligible, given the nonlinearity $g(u_{ij}(t))$. Thus, over one sawtooth period, weights updates $\Delta w_{ij}$ are proportional to the contrastive signal $s_{ij}^{\text{clamped}} - s_{ij}^{\text{free}}$.

Intuitively, the kernel $K$ computes the approximate (finite) time derivative of $s_{ij}(t)$; this is what requires the kernel timescale $\tau_K \ll \tau_f$ (Fig. S1). The rapid rise of the sawtooth from free to clamped states results in a large derivative that exceeds the threshold $\theta_g$ in $g(u)$ and provides the necessary contrastive learning update. The slow relaxation from clamped to free has a small time-derivative that is below $\theta_g$. The ability to distinguish free-to-clamped versus clamped-to-free transitions requires $\tau_f \ll \tau_s$ as shown in Section S1.

Consequently, the fast-slow sawtooth protocol allows systems with non-equilibrium memory kernels to naturally learn through contrastive rules by comparing free and clamped states over time. Note that our model here does not include an explicit memory module that stores the free and clamped state configurations but rather exploits the memory implicit in local feedback dynamics at each synapse.

### Related work summary

Several works have proposed ways in which contrastive learning can be generated in natural and engineered systems, such as: utilizing explicit memory[27,28]; having two globally coordinated phases each with low learning rates[46]; having two copies of the system[47–49]; having two physically distinct kinds of signals[15]; or using continually-running oscillations in the learning rules[16,24]. All these methods require globally switching the system between two phases or exploit hardware-specific mechanisms. The closest to the TCL approach here is 'continual EP'[50], but it also requires globally switching between phases of learning. See Section S2 for a more in-depth review of these related works.

Our rule superficially resembles spike-timing dependent plasticity (STDP)[42,51,52], in particular how our TCL weight update has a representation involving a non-monotonic kernel $K$. While STDP-inspired rules[53–55] and other rules involving competing Hebbian updates with inhibitory neurons[56,57] have shown great promise as local learning paradigms, they are adapted to settings where synapses are asymmetric and can distinguish differentials between the timings of pre- and post-synaptic neuronal activations. Our rule only involves signals at the same moment in time $t$ and can only result in symmetric interactions $w_{ij}$. We are not aware of any direct relationship between work on STDP rules and the proposal here; see Section S2 for further detail.

### Performance on MNIST

By coupling together a network of synapses with non-equilibrium memory kernels, we were able to train a neural network capable of classifying MNIST. In particular, we adapted the Equilibrium Propagation (EP) algorithm[26], a contrastive learning-based method that 'nudges' the system's state towards the desired state, rather than clamping it as is done in standard contrastive Hebbian learning (CHL)[22]. We used EP instead of CHL due to its better properties and its superior performance in practice[58,59]. EP makes weight updates of the form:

$$\Delta w_{ij} = \epsilon \left( s_{ij}^{\text{nudge}} - s_{ij}^{\text{free}} \right). \tag{8}$$

Normally the EP method requires storing the states $s_{ij}^{\text{nudge}}$ and $s_{ij}^{\text{free}}$ in memory, computing the difference and then updating weights $w_{ij}$; our proposed TCL method will accomplish the above EP weight update, without explicitly storing and retrieving those states.

In order to process MNIST digits, we utilize a network architecture with three types of symmetrically-coupled nodes. Each node carries internal state $x$ and activation $\eta = \text{clip}(x, 0, 1)$. Nodes belong either to a 784-node input layer (indexed by $i$), a 500-node hidden layer (indexed by $h$), or a 10-node output layer (indexed by $o$). Nodes are connected by synapses only between adjacent layers ($i$ and $h$, $h$ and $o$), with no skip- or lateral-layer couplings. The neural network dynamics minimize the energy:

$$E(x) = \frac{1}{2}\Sigma_n x_n^2 - \frac{1}{2}\Sigma_{n,m} w_{nm}\eta_n\eta_m - \Sigma_n b_n\eta_n, \tag{9}$$

where $b_n$ is the bias of node $n$, $w_{nm}$ is the weight of the synapse connecting nodes $n$ and $m$, and the indices $n, m$ run over the node indices of all layers $i, h, o$.

We represent each 784-pixel grayscale MNIST image as a vector $v^{\text{image}} = \{v_i^{\text{image}}\}_{i=0,\ldots,783}$. For each MNIST digit $v^{\text{image}}$, we hold the states of the 784 input nodes at constant values over time, $x_i(t) = v_i^{\text{image}}$. For inference, we allow the hidden- and output-layer activations to adjust in response to the fixed input nodes, minimizing Eq. (9). Once a steady-state is reached, the network prediction is given by looking at the states of the 10 output nodes, $\{x_o\}_{o=0,\ldots,9}$, and interpreting the index of the maximally activated output node as the input image label.

During inference, the network minimizes an energy function which does not vary in time, subject to the constraint that $x_i(t) = v_i^{\text{image}}$. During training, the same constraint $x_i(t) = v_i^{\text{image}}$ applies, but our network instead is subjected to a time-varying energy function:

$$F(x;t) = E(x) + \frac{\beta(t)}{2}\Sigma_{o=0}^9 \left( x_o - v_o^{\text{label}} \right)^2. \tag{10}$$

Here, $v^{\text{label}}$ is the one-hot encoding vector for the corresponding label of the MNIST digit. The time-dependent training protocol $\beta(t)$ is the asymmetric sawtooth function which smoothly interpolates between 0 and a maximal value $\beta_{max} < 1$. One portion of the sawtooth is characterized by the fast timescale $\tau_f$, and the other portion by the slow timescale $\tau_s$. We assume that both these timescales are quasi-static compared to any system-internal energy relaxation timescales[60]. This restriction to adiabatic protocols is a potential limitation to contrastive methods in general; details depend on the system and some works present viable workarounds[25,61].

Each time the training protocol $\beta(t)$ completes a cycle, the network weights are updated according to Eq. (6), with $\eta_n\eta_m$ as the necessary synaptic current $s_{nm}$. After completing multiple sawtooth cycles, we switch the inputs $x_i$ to a new MNIST image and repeat the training process of manipulating the $x_o$ through the time-varying energy function $F(x; t)$. See Section S4 for further detail, including specifications of $K$ and $g$ for Eq. (6).

We find that, after training for 35 epochs, our classification error drops to 0, and we achieve an accuracy of 95% on our holdout test dataset (Fig. 3B). Our results demonstrate the feasibility of performing contrastive learning in neural networks without requiring explicit memory storage, but leave open the question of limitations on our approach. We consider performance limitations at the level of a single synapse in the following sections.

### Speed-accuracy tradeoff

The learning model proposed here relies on breaking the symmetry between free and clamped using timescales in a sawtooth protocol (Fig. 2C). The expectation is that our proposal for approximating the difference between free and clamped states works best when this symmetry-breaking is large; i.e. in the limit of slow protocols.

We systematically investigated such speed-accuracy trade-offs inherent to the temporal strategy proposed here. First, we fix synaptic parameters $\theta_g, \tau_K$ and protocol parameters ($\tau_s, \tau_f$) and consider protocols of varying amplitude $A = s^{\text{clamped}} - s^{\text{free}}$ (Fig. 4A). The protocol we use can be written in the form:

$$s_{ij}(t) = \begin{cases} \frac{At}{\tau_f} + \overline{s_{ij}} - \frac{A}{2} & t \le \tau_f \\[2mm] \overline{s_{ij}} + \frac{A}{2} - \frac{A(t-\tau_f)}{\tau_s} & \tau_f < t \le \tau_s \end{cases}, \tag{11}$$

with $\overline{s_{ij}}$ the average of the clamped and free states. We fix the kernel $K$ to be of the form $K(t-t') \sim e^{-(t-t')/\tau_K}f(t-t')$, with the exact expression for the polynomial $f$ given in Section S5.

We compute $\Delta w_{ij}$ for these protocols and plot against amplitude $A$ in Fig. 4B. A line with zero intercept and slope 1 indicates a

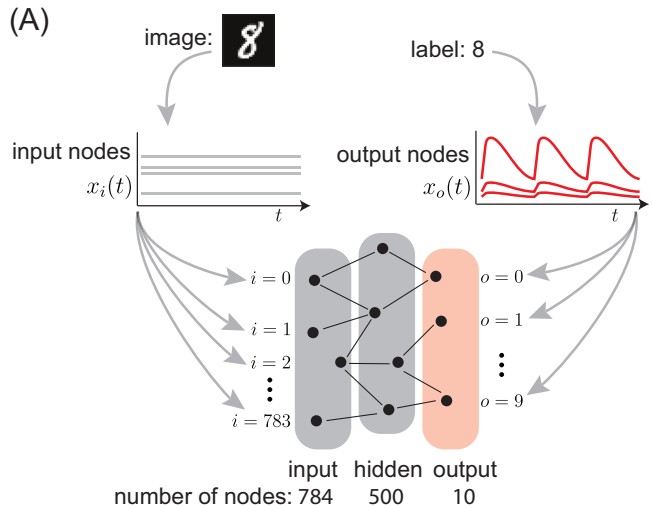

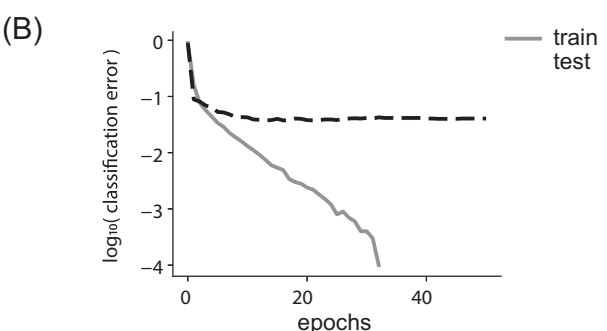

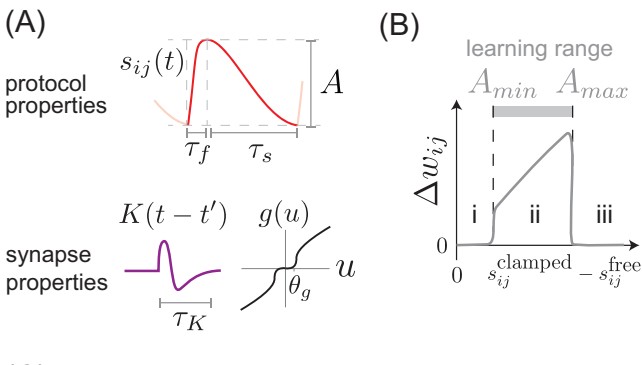

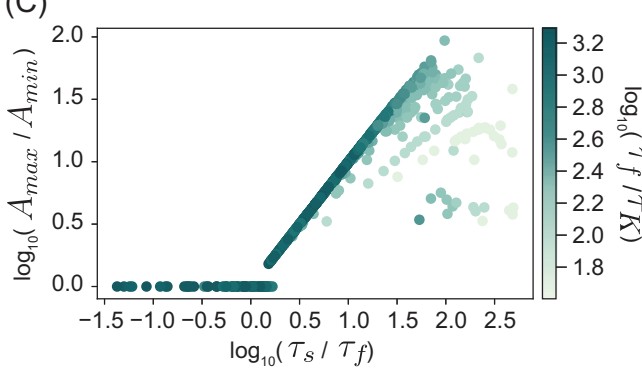

**Fig. 4 | Tradeoff between time and dynamic range of synaptic currents over which contrastive learning is approximated. A** The sawtooth training protocol $s_{ij}(t)$ with amplitude $A$ approximates contrastive learning in specific regimes of protocol timescales ($\tau_s$, $\tau_f$) relative to nonlinearity threshold $\theta_g$ and synaptic memory kernel timescale $\tau_K$. **B** The weight update $\Delta w_{ij}$ is proportional to the signal amplitude $s^{\text{clamped}} - s^{\text{free}}$ as desired within a range of amplitudes (regime ii) for given timescales. **C** Learning range (defined in (**B**) as the dynamic range $A_{max}/A_{min}$ for which the sawtooth protocol approximates contrastive Hebbian learning) plotted against $\tau_s/\tau_f$. Signal timescales ($\tau_s$, $\tau_f$) place a limit on maximal learning range which increases with increasing $\tau_s/\tau_f$. Protocols with higher $\tau_f/\tau_K$ approach this limit.

---

**Fig. 3 | Temporal Contrastive Learning in a neural network for MNIST classification. A** We train a neural network with three types of neurons: input, hidden, and output. During training time, neurons in the input layer are set to internal states $x_i$ based on input MNIST digits (gray curves). But output neuron states $x_o$ are modulated between being nudged to the desired output (i.e., correct image label) and being free in a sawtooth-like protocol (red curves). Synaptic weights are updated using a memory kernel with timescale $\tau_K = .1$; length of one free-clamped cycle $\tau_f + \tau_s = 1$, $\tau_f = .1$. See Section S4 for protocol details. **B** Average train classification error as a function of epochs of training (gray curve). Each epoch involves a pass through the 10000 MNIST entries in the training set, where for each MNIST entry, network weights are updated based on a single cycle of the sawtooth protocol. Test error (dashed black curve) is computed on 2000 entries in a test set.

$s_{ij}(t)$. As $\tau_f \rightarrow \tau_K$, the contrastive Hebbian approximation breaks down (see Section S1). By setting $\tau_f/\tau_K$ to be just large enough to capture the derivative of $s_{ij}(t)$, we find that the central trade-off to be made is between having a longer $\tau_s$ and a larger dynamic range $A_{max}/A_{min}$ (Fig. 4C).

## Energy dissipation cost of non-equilibrium learning

Our mechanism is fundamentally predicated on memory as encoded by the non-monotonic kernel $K(t - t')$. At a fundamental level, such memory can be linked to non-equilibrium dynamics to provide a Landauer-like principle for learning[31,62]. Briefly, we will show here that learning accuracy is reduced if the kernel has non-zero area $I = \int_0^\infty K(t)dt$; we then show, based on prior work[31,63], that reducing the area of kernels to zero requires increasingly large amounts of energy dissipation. In other words, to perform increasingly accurate inference, systems need to consume more energy (e.g. electrical energy in neuromorphic systems, ATP or other chemical fuel in molecular systems), which is then dissipated as heat.

We first consider the problem at the phenomenological level of kernels $K$ with non-zero integrated area $I = \int_0^\infty K(t)dt$ (Fig. 5A, Section S6). In the limit of small $I$ and $\tau_K$, and large $\tau_s$, the weight update $\Delta w$ to a synapse experiencing a current $s_{ij}(t)$ reduces to:

$$\epsilon^{-1}\Delta w_{ij} = \left(s_{ij}^{\text{clamped}} - s_{ij}^{\text{free}}\right) + \frac{\tau_f I}{2}\left(s_{ij}^{\text{free}} + s_{ij}^{\text{clamped}}\right) \quad (13)$$

perfect contrastive Hebbian update. We see that the contrastive Hebbian update is approximated for a regime of amplitudes $A_{min} < A < A_{max}$, where $A_{min}$, $A_{max}$ are set by the requirement that $\theta_g$ separates the rate of change in the fast and slow sections of the protocol (Fig. 4B).

We determined this dynamic range $A_{max}/A_{min}$ for protocols of different timescales $\tau_s$, $\tau_f$, keeping the kernel fixed but always choosing a threshold $\theta_g$ which provides an optimal dynamic range for the synapse output, given a maximal amplitude magnitude (see Section S1). Choosing $\theta_g$ this way, we find the following trade-off equation,

$$\tau_s > \tau_f \frac{A_{max}}{A_{min}}, \quad (12)$$

that is, the slower the speed of the down ramp, the larger the range of amplitudes over which contrastive learning is approximated (Fig. 4C).

Finally, we also need $\tau_K \ll \tau_f$; intuitively, the kernel $K(t - t')$ is a derivative operator but on a finite timescale $\tau_K$; thus the fast ramp must last longer than this timescale so that $u_{ij}(t)$ can reflect the derivative of

(A) synapse properties     (B)

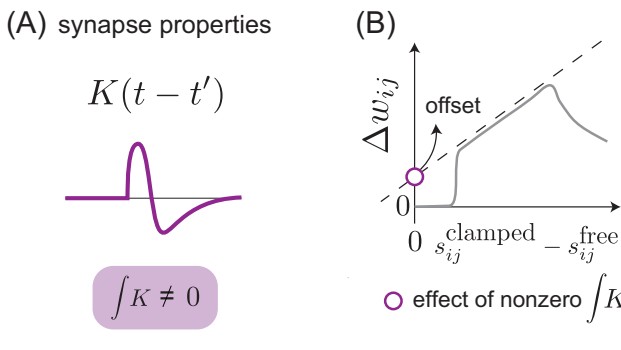

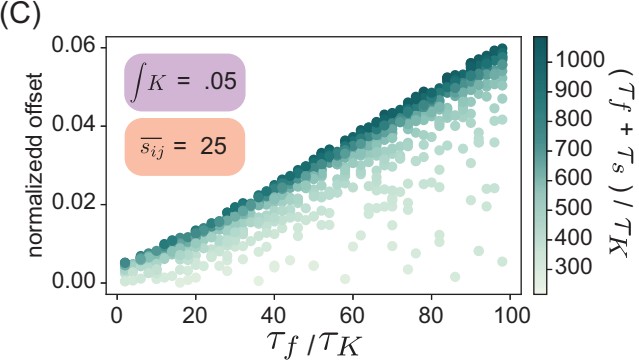

**Fig. 5 | Kernels with nonzero area limit contrastive learning. A** Memory kernels with nonzero area ($I = \int_0^\infty K dt$) are sensitive to the time-average synaptic current $\overline{s_{ij}}$. **B** For protocols with kernels with nonzero area, $\Delta w_{ij}$ is no longer proportional to $s^{\text{clamped}} - s^{\text{free}}$ within the range of performance but instead has a constant offset. Offset is the y-intercept of the line of best fit for the linear portion of the weight update curve. **C** Offset from contrastive learning (defined in (**B**)) for protocols of varying $\tau_f$, $\tau_s$, evaluated for a fixed memory kernel of timescale $\tau_K$ and non-zero area. Offset is positively correlated with $\tau_f$, with overall longer protocols (high $\tau_f + \tau_s$) offset the most. Here the average signal value $\overline{s_{ij}} = 25$; offset is normalized by $\overline{s_{ij}}$.

where $\tau_f$ is the fast timescale of the protocol. See Section S6 for further detail. Hence, the response of the synapse with a non-zero area deviates from the ideal contrastive update rule by the addition of an offset proportional to the average signal value (Fig. 5B).

Indeed, when we fix integrated kernel area and mean signal value, we find that increasing protocol lengths (both $\tau_f$ and $\tau_s$, due to finite $\tau_K$) result in a larger offset, and hence further deviations from the ideal contrastive update (Fig. 5C). This constraint is in contrast to the metric used for assessing performance of kernels with zero integrated area, where we found that longer protocols enabled learning over a wider dynamic range of amplitudes. This result suggests that optimal protocols will be set through a balance between the desire to minimize the offset created by non-zero kernel area, and the desire to maximize the dynamic range of feasible contrastive learning.

We now relate the kernel area to learning accuracy to provide a Landauer-like[62] relationship between energy dissipation and contrastive learning. To understand this fundamental energy requirement, we consider a microscopic model of reversible non-equilibrium feedback. Such models have been previously studied to understand the fundamental energy cost of molecular signal processing during bacterial chemotaxis[31,64]. In brief, non-monotonic memory kernels $K$ require breaking detailed balance and thus dissipation; further, reducing the area $I$ to zero requires increasingly large amounts of dissipation.

As shown in Refs. 31,64, the simplest way to model a non-monotonic kernel with a fully reversible non-equilibrium statistical model is to use a Markov chain shaped like a ladder network (Section S8, Fig. S2). The dynamics of the Markov chain are governed by the

master equation:

$$\frac{d}{dt}p_a = \sum_b r_{ba}p_b - p_a \sum_b r_{ab}. \tag{14}$$

Here $r_{ab}$ are the rate constants for transitions from state $a$ to $b$, and $p_a$ is the occupancy of state $a$. We consider a simple Markov chain network arranged as a grid with two rows (see Fig. 6A for a schematic, and Section S8 for the fully specified network). The dynamics of the network are mainly controlled by two parameters: $s_{ij}$ and $\gamma$. As in Ref. 64, the rates ($r_{up}^i$, $r_{down}^i$) on the vertical (red) transitions are driven by synaptic current $s_{ij}$. These rates vary based on horizontal position such that circulation along vertical edges is an increasing function of $s_{ij}$; see Section S8 for the functional form of this coupling. Quick changes in $s_{ij}$ shift the occupancy $\vec{p}$, which then settles back into a steady state. Thus through $s_{ij}$ the network is perturbed by an external signal. On the other hand, the parameter $\gamma$ controls the intrinsic circulation within the network by controlling the ratio of clockwise ($r_{cw}$) to counterclockwise ($r_{ccw}$) horizontal transition rates:

$$\gamma = \frac{r_{cw}}{r_{ccw}}. \tag{15}$$

Then $u_{ij}$ is taken to be the occupancies $\sum_i p_i$ over all nodes $i$ along one rail of the ladder (Fig. 6A). The parameter $\gamma$ is particularly key in quantifying the breaking of detailed balance. When $\gamma \to 0$, i.e., when this network is driven out of equilibrium, the microscopic dynamics of this Markov chain model provide a memory kernel $K$ suited for contrastive learning. In particular, the response of the Markov chain to a small step function perturbation in $s_{ij}$ (i.e., to vertical rate constants) results in $u_{ij}(t) = \int_{-\infty}^t K(t - t')s_{ij}(t')dt'$ with a memory kernel $K(t - t')$ much as needed for Fig. 2B. The form of $K$ can then be extracted from $u_{ij}$ by taking its derivative and normalizing.

In the Markov chain context, we can rigorously compute energy dissipation $\sigma = \sum_{i > j} (r_{ij}p_j - r_{ji}p_i) \ln(\frac{r_{ij}p_j}{r_{ji}p_i})$.

For the particular ladder network studied here, dissipation takes the following form (in units of $kT$):

$$\sigma = \sigma_v + \sum_{i=0}^8 (p_i - \gamma p_{i+1}) \ln\left(\frac{p_i}{\gamma p_{i+1}}\right), \tag{16}$$

where $\sigma_v$ is dissipation along vertical connections, and the node occupations $p_i$ (excluding corners, which are accounted for by $\sigma_v$) are indexed in counterclockwise order. The dissipation is a decreasing function of $\gamma \in [0, 1]$. This dissipation (at steady state), is zero if $\gamma = 1$ (i.e., detailed balance is preserved) and non-zero otherwise. Markov chains with no dissipation produce monotonic memory kernels which do not capture the finite time derivative of the signal.

However, for a finite amount of non-equilibrium drive $\gamma$, the generated kernel will generally have both a positive and negative lobe and have lower but non-zero integrated area $I$, leading to imperfect implementation of the contrastive learning rule, offset as a result of non-zero area. Larger dissipation of the underlying Markov chain, e.g., by decreasing $\gamma \to 0$, will lower the learning offset and cause the kernel convolution to approach an exact finite time derivative. Figure 6B shows the increased dissipation required to reach lower offset.

In summary, Landauer's principle[62] relates the fundamental energy cost of computation to the erasure of information inherent in most computations. Our work here provides a similar rational for why a physical system learning from the environment must similarly dissipate energy - contrastive learning fundamentally requires comparing states seen across time. Hence such learning necessarily requires temporarily storing information about those states and erasing that information upon making weight updates. While the actual energy dissipation can vary depending on implementation details in a real

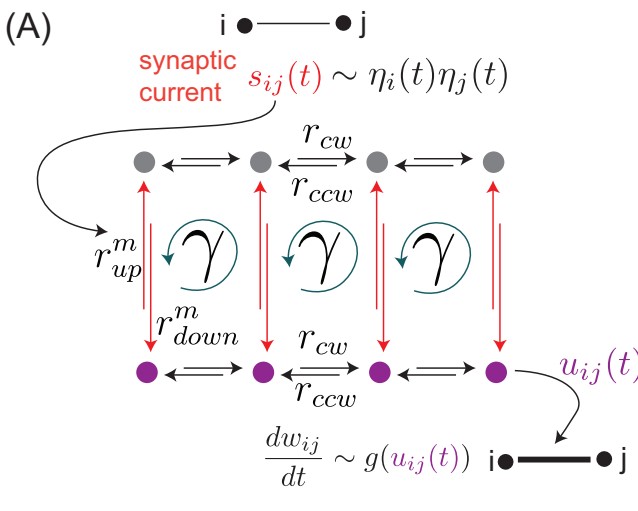

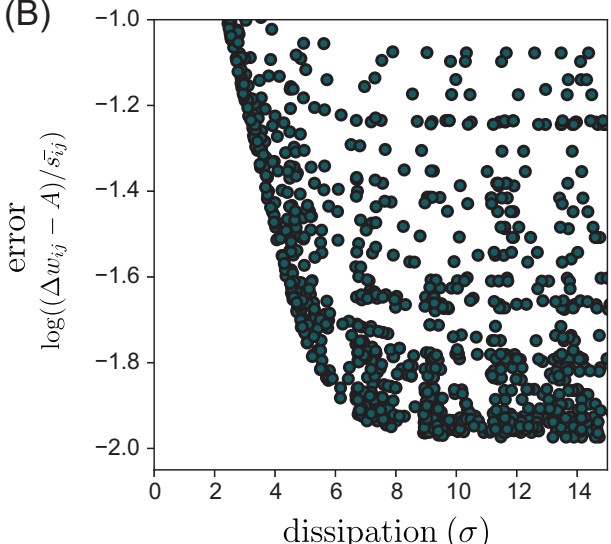

**Fig. 6 | Dissipation cost of contrastive learning. A** A thermodynamically reversible Markov state model of physical systems with non-monotonic kernel $K$. Nodes represent states connected by transitions with rate constants as shown; the ratio of rate constants $r_{up}^m$ and $r_{down}^m$ is set by synaptic current $s_{ij}(t)$ (the input). The output $u_{ij}$ is defined as the resulting total occupancy of purple states; $u_{ij}$ is used to update weight $w_{ij}$. $g$ is a threshold nonlinearity as in Fig. 4A. The ratio of products of clockwise to counterclockwise rate constants, $\gamma$, quantifies detailed balance breaking and determines energy dissipation $\sigma$. **B** Energy dissipation $\sigma$ versus error in contrastive learning rule for different choices of rate constants in (**A**). Error of contrastive learning is quantified by the normalized offset from ideal contrastive weight update; $\Delta w_{ij}$ is the weight update $\int_0^t g(u(t'))dt'$, $A$ is the signal amplitude $s_{ij}^{\text{clamped}} - s_{ij}^{\text{free}}$, and $\bar{s}_{ij}$ is the average signal value $(s_{ij}^{\text{clamped}} + s_{ij}^{\text{free}})/2$ for a sawtooth signal (see Fig. 5B). Bounded region indicates greater dissipation is required for lower offset and better contrastive learning performance.

system, our kernel-based memory model allows for calculating this dissipation cost associated with learning in a reversible statistical physics model.

**Realizing memory kernels through integral feedback**
We have established that contrastive learning can arise naturally as a consequence of non-equilibrium memory as captured by a non-monotonic memory kernel $K$. Here, we argue that the needed memory kernels $K$ in turn arise naturally as a consequence of integral feedback control in a wide class of physical systems. Hence many simple physical and biological systems can be easily modified and manipulated to undergo contrastive learning.

Integral feedback control[30,65–67] is a broad homeostatic mechanism that adjusts the output of a system based on measuring the integral of error (i.e., deviation from a fixed point). For example, consider a variable $u$ that must be held fixed at a set point $u_0$ despite perturbations from a signal $s$. Integral feedback achieves such control by up- or down-regulating $u$ based on the integrated error signal $u - u_0$ over time,

$$\tau_u \frac{du}{dt} = -u + k(s(t) - m) \tag{17}$$

$$\tau_m \frac{dm}{dt} = (u - u_0) \tag{18}$$

Here, we rely on a well-known failing of this mechanism[30,31] - if $s(t)$ changes rapidly (e.g., a step function), the integral feedback will take a time $\tau_K$ to restore homeostasis; in this time, $u(t)$ will rise transiently and then fall back to $u_0$ as shown in Fig. 2B. The general solution of Eqs. (17), (18) is of the form of a driven damped oscillator driven by the time derivative of $s(t)$[29,30]. The kernel $K(t)$ can be written as the solution for forcing $s(t) = \delta(t)$. In the overdamped regime $k\tau_u \ll \tau_m$, the kernel is of the form $K(t) = e^{-bt}\left(\cosh(\omega t) - \frac{b}{\omega}\sinh(\omega t)\right)\Theta(t)$, where $b = \frac{1}{2\tau_u}$, $\omega = \sqrt{\frac{k}{\tau_u \tau_m} - \frac{1}{4\tau_u^2}}$, $\Theta$ is a unit step function, and we have chosen $u_0 = 0$. This form of $K(t)$ is a non-monotonic function with both a positive and negative lobe, which returns to the set point $u_0 = 0$ after a perturbation. Hence simple integral feedback can create memory kernels of the type required by this work. We propose a large class of physical systems capable of Hebbian learning can be promoted to contrastive learning by a layer of integral feedback as shown Fig. 7A.

**Learning in mechano-chemical systems.** Numerous examples of biological networks release chemical signals in response to mechanical deformations $s_{ij}$; e.g., due to shear flow forces in *Physarum polycephalum*[68–70] or strain in cytoskeletal networks[71,72]. If the chemical signals then locally modify the elastic moduli or conductances of the network, these systems can be viewed as naturally undergoing Hebbian learning.

Our proposal suggests that these systems can perform contrastive learning if the chemical signal is integral feedback regulated (Fig. 7B). For example, if the molecule released by the mechanical deformation is under negative autoregulation, then that molecule's concentration $u_{ij}(t)$ is effectively the time derivative of the mechanical forcing of the network, $\dot{s}_{ij}(t)$. A simple model for negative autoregulation is:

$$\tau_u \frac{du}{dt} = -u + k_a s(t) - k_i m \frac{u}{u_s + u} \tag{19}$$

$$\tau_m \frac{dm}{dt} = u - u_0, \tag{20}$$

Here, $u$ is the level of activated molecules (e.g., phosphorylated form); we assume that phosphorylation is driven with rate $k_a$ by the strain molecule $s$. However, excess levels of $u$ relative to a baseline $u_0$ leads to build up of another molecular form $m$ (e.g., methylated molecules in the case of chemotaxis[29,73]). If we assume that $m$ then deactivates (or dephosphorylates as in chemotaxis) $u$ with rate $k_i$, we obtain an control loop as long as $u$ remains above a small saturating threshold $u_s$. These dynamics provide integral feedback control over $u$[74], and allow for computing the time-derivative of strain $s$.

Therefore, altering the radius in flow networks or stiffness in elastic networks based on $u_{ij}(t)$ will now lead to contrastive learning. The same mechano-chemical principles can also be exploited in engineered systems such as DNA-coupled hydrogels[75]; in these systems, DNA-based molecular circuits[76–79] can implement the needed

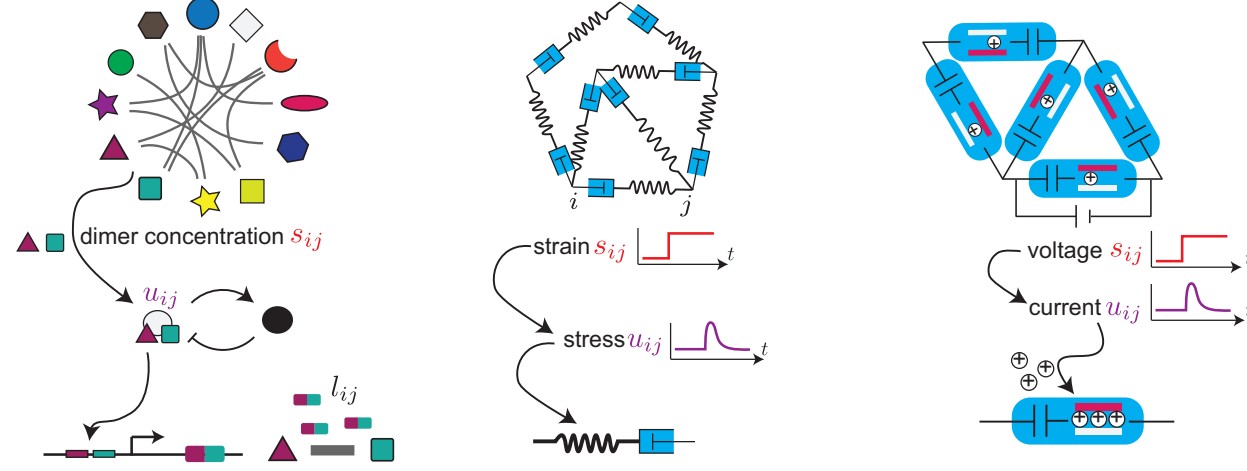

**Fig. 7 | Integral feedback allows diverse Hebbian-capable systems to achieve contrastive learning. A** Synaptic current $s_{ij}$ in synapse $ij$ impacts a variable $u_{ij}(t)$ which is under integral feedback control. That is, the deviation of $u_{ij}(t)$ from a setpoint $u_0$ is integrated over time and fed back to $u_{ij}$, causing $u_{ij}(t)$ to return to $u_0$ after transient perturbations due changes in $s_{ij}(t)$. We can achieve contrastive learning by updating synaptic weights $w_{ij}$ using $u_{ij}(t)$, as opposed to Hebbian learning by updating $w_{ij}$ based on $s_{ij}(t)$. **B** Mechanical or vascular networks can undergo Hebbian learning if flow or strain produce molecular species (blue hexagon $u_{ij}$) that drive downstream processes that modify radii or stiffnesses of network edges. But if those blue molecules are negatively autoregulated (through the green species as shown), these networks can achieve contrastive learning. **C** Molecular interaction networks can undergo Hebbian learning if the concentrations of dimers

$s_{ij}$, formed through mass-action kinetics, drives expression of linker molecules $l_{ij}$ (here, purple-green rectangles) that mediate binding interactions between monomers $i, j$. But if transcriptionally active dimers $u_{ij}$ additionally stimulate regulatory molecules (black circle) which inhibit $u_{ij}$, then levels of linker molecules $l_{ij}$ will provide interactions learned through contrastive rules. **D** Stresses in networks of viscoelastic mechanical elements (dashpots connected in series to springs) reflect the time-derivatives of strains due to relaxation in dashpots. These stresses can be used to generate contrastive updates of spring stiffnesses. **E** Nanofluidic memristor networks can undergo Hebbian updates by changing memristor conductance in response to current flow. However, if capacitors are added as shown, voltage changes $s_{ij}$ across a memristor results in a transient current $u_{ij}$; conductance changes due to these currents $u_{ij}$ result in contrastive learning.

integral feedback while interfacing with mechanical properties of the hydrogel. Such chemical feedback could be valuable to incorporate into a range of existing metamaterials with information processing behaviors[18,80].

**Learning in molecular systems.** Molecular systems can learn in a Hebbian way through 'stay together, glue together' rules, analogous to the 'fire together, wire together' maxim for associative memory in neural networks[6–10,81]. In the example shown in Fig. 7C, temporal correlations between molecular species $i$ and $j$ would result in an increased concentration of linker molecules $l_{ij}$ that will enhances the effective interaction between $i$ and $j$. This mechanism exploits dimeric

transcription factors[82,83]; an alternative proposal[8] involves proximity-based ligation[84] as used in DNA microscopy[85].

We can promote these Hebbian-capable systems to contrastive learning-capable systems by including a negative feedback loop. In this scheme, monomers $i, j$ form dimers $s_{ij} \propto c_i c_j$ dictated by mass-action kinetics, as in Hebbian molecular learning[6,8]. We now assume that these dimers can form a transcriptionally active component $u_{ij}$ by binding an activating signal (e.g. shown as a circle in Fig. 7C).

Here, molecular learning[6,8] takes place as follows: monomers $i, j$ form a compound dimer $u_{ij}$ whose concentration is dictated by mass-action kinetics $s_{ij} \propto c_i c_j$; these $u_{ij}$ drive the transcription of a linker molecule $l_{ij}$ that mediates Hebbian-learned interactions between $i$ and

*j*. To achieve Temporal Contrastive Learning, we now additionally allow activated dimers $u_{ij}$ to produce a regulating signal $m_{ij}$ which deactivates or degrades $u_{ij}$ during training time[82,86]. At test time, the resulting interaction network for monomers *i,j*, mediated by linker levels $l_{ij}$, will reflect interactions learned through contrastive rules.

This scheme can be quantitatively written as:

$$\tau_u \frac{du}{dt} = -u + k_a s(t) - k_i m \frac{u}{u_s + u} \qquad (21)$$

$$\tau_m \frac{dm}{dt} = u - u_0 \qquad (22)$$

$$\tau_l \frac{dl}{dt} = g(u - u_0), \qquad (23)$$

where $k_a$ and $k_i$ are rates of production and degradation, $u_s$ is a small saturating threshold for degradation, $u_0$ is a baseline for the production of *m*, and *g* is a transcription-related nonlinearity for the production of linkers *l*.

**Learning in viscoelastic materials.** Mechanical materials with some degree of plasticity have frequently been exploited in demonstrating how memory formation and Hebbian learning can arise in simple settings, e.g. with foams[11,87], glues[18], and gels[88]. In this setting, stress slowly softens the moduli of highly strained bonds, thereby lowering the energy of desired material configurations in response to a strain signal $s_{ij}(t)$.

We add a simple twist to this Hebbian framework in order to make such systems capable of contrastive learning; we add a viscoelastic element with a faster timescale of relaxation compared to the timescale of bond softening. In reduced-order modeling of viscoelastic materials, a common motif is that of a viscous dashpot connected to an elastic spring in series, also known as a Maxwell material unit. Such units obey the following simple equation:

$$\frac{\dot{u}}{k} + \frac{u}{\gamma} = \dot{s} \qquad (24)$$

where *u* is the stress in the unit, *s* is the strain in the unit, *k* is the Hookean modulus of the spring and $\gamma$ is the Newtonian viscosity. In this simple model, a Maxwell unit experiences integral feedback; sharp jumps in the strain $s_{ij}(t)$ lead to stresses $u_{ij}(t)$ across the edge which initially also jump, but then relax as the viscous dashpot relieves the stress. As such, the stress in each edge of the network naturally computes an approximate measurement of the time derivative of strain the edge experiences, $\dot{s}_{ij}(t)$. We can see this cleanly by solving for the case where $s(t) = \dot{s}t$ with $\dot{s}$ constant, in which case $u(t) = \gamma \dot{s}(1 - e^{-\frac{k}{\gamma}t})$, in which case $u \propto \dot{s}$ after a relaxation time $\frac{\gamma}{k}$.

We consider a network of such units (Fig. 7D), experiencing strains $s_{ij}(t)$ across each edge. If the springs in this network naturally adapt their moduli $k_{ij}$ on a slow timescale in response to the stresses they experience, they would naturally implement Hebbian learning rules. With the introduction of viscoelasticity through the dashpots, the system is now capable of contrastive updates. Note that, in contrast to the solution from idealized Eqs. (17), (18), the material moduli $k_{ij}$ are involved both as learning degrees of freedom and as elements of the integral feedback control.

**Learning in nanofluidic systems.** An emerging platform for neuromorphic computing involves nanofluidic memristor networks (Fig. 7E) driven by voltage sources[89–92]. Each memristic element naturally displays Hebbian behavior; as voltage drops across the memristor, ions are recruited which modify the conductance of the element. In this Hebbian picture, the voltage drop across a network element plays the role of the synaptic current $s_{ij}$.

Here, we make a simple modification to allow for contrastive learning, where each element in the network is composed not of a single memristor but of a capacitor and memristor connected in series. If voltage drops are established across a compound capacitor memristor element, then voltage initially drives a current across the memristor. However, this initial spike in current is suppressed on the timescale of the charging capacitor. To see this, note that as in the viscoelastic system (Eq. (24)), each unit has dynamics of the form:

$$R\dot{u} + \frac{u}{C} = \dot{s}, \qquad (25)$$

where here *u* is the current in the unit, *s* is the voltage across the unit, *R* is the resistance and *C* is the capacitance. For the case where $\dot{s}$ is constant, $u(t) = C\dot{s}(1 - e^{-\frac{t}{RC}})$, in which case $u(t) \propto \dot{s}$ after a relaxation time *RC*.

Therefore, due to the feedback control from the capacitor, the voltage that the memristor develops is $u \approx \dot{s}$. This simple modification therefore naturally allows for nanofluidic systems which update network conductances contrastively.

## Discussion

Backpropagation is a powerful way of training neural networks using GPUs. Training based on local rules – where synaptic connections are updated based on states of neighboring neurons – offer the possibility of distributed training in physical and biological systems through naturally occurring processes. However, one powerful local learning framework, contrastive learning, seemingly requires several complexities; naively, contrastive learning requires a memory of 'free' and 'clamped' states seen over time and/or requires the learning system to be globally switched between Hebbian and anti-Hebbian learning modes over time. Here, we showed that such complexities are alleviated by exploiting non-equilibrium memory *implicit* in the integral feedback update dynamics at each synapse that is found in many physical and biological systems.

Our Temporal Contrastive Learning approach offers several conceptual and practical advantages. In comparison to backpropagation, it provides a learning algorithm in hardware where no central processor is available. In comparison to other local learning algorithms, which may still require some digital components, our approach offers several advantages. To start, TCL does more with less. A single analog operation − Hebbian weight update based on an integral feedback-controlled synaptic current − effectively stores memory of free and clamped states, retrieves that information, computes the difference, and updates synaptic weights. No explicit memory element is needed. Further, since integral feedback occurs naturally in a range of systems[29,86] and we exploit a failure mode inherent to integral feedback, our approach can be seen as an example of the non-modular 'hardware is the software' philosophy[93] that provides more robust and compact solutions.

Our framework has several limitations quantified by speed-accuracy and speed-energy tradeoffs derived here, in addition to the general concerns about adiabatic protocols in contrastive methods[25,60]. The time costs in our scheme (e.g. those shown in Fig. 4) are inherently larger compared to other Equilibrium Propagation-like methods which use explicit memory and global signals to switch between wake and sleep, thereby avoiding slow ramping protocols. Slower training protocols in systems with higher energy dissipation lead to better approximations of true gradient descent on the loss function. However, we note that *not* performing true gradient descent is known to provide inductive biases with generalization benefits in other contexts[94,95]; we leave such an exploration to future work.

Darwinian evolution is the most powerful framework we know that drives matter to acquire function by experiencing examples of such function over its history[96]; however, Darwinian evolution requires self-

replication. Recent years have explored 'physical learning'[5] as an alternative (albeit less powerful) way for matter to acquire functionality without self-replication. For example, in one molecular version[6,7], recently realized at the nanoscale[8], molecules with Hebbian-learned interactions can perform complex pattern recognition on concentrations of hundreds of molecular species, deploying different molecular self-assembly on the nanoscale. Similar Hebbian-like rules have allowed physical training of mechanical systems[11,18,97] for specific functionality. This current work shows that natural physical systems can exploit more powerful learning frameworks that require both Hebbian and anti-Hebbian training[13,14] by exploiting integral feedback control. Since integral feedback is relatively ubiquitous and achieved with relatively simple processes, the work raises the possibility that sophisticated non-Darwinian learning processes could be hiding in plain sight in biological systems. In this spirit, it is key to point out that other ideas for contrastive learning have been proposed recently[15,16]. Further, the contrastive framework is only one potential approach to learning through local rules plausible in physical systems; the feasibility of other local frameworks[57,98,99] in physical systems remain to be explored.

By introducing a reversible statistical physics model for the memory needed by contrastive learning, we provided a fundamental statistical physics perspective on the dissipation cost of learning. While specific implementations will dissipate due to system-specific inefficiencies, our work points at an unavoidable reason for dissipation in the spirit of Landauer[62]. Our analysis of dissipation here only focuses on the contrastive aspect of contrastive learning. Developing reversible models for other aspects of the learning process, as has been done for inference and computation extensively[31,62,100–107], can shed light on fundamental dissipation requirements for learning in both natural and engineered realms.

## Methods

### Temporal Contrastive Learning weight updates
In Temporal Contrastive Learning, we update weights following Eq. (6):

$$\Delta w_{ij} = \epsilon \int_0^{\tau_f + \tau_s} g\left(\int_{-\infty}^t K(t - t') s_{ij}(t') \, dt'\right) dt, \tag{26}$$

with $\epsilon$ the learning rate. The temporal forcing from the synaptic current $s_{ij}(t)$ is assumed to be an asymmetric sawtooth wave: a fast transition from the free value $s_{ij}^{\text{free}}$ to the clamped value $s_{ij}^{\text{clamped}}$ occurring over a timescale $\tau_f$, and then a slow relaxation from clamped back to free over a timescale $\tau_s$:

$$s_{ij}(t) = \begin{cases} \frac{At}{\tau_f} + \overline{s_{ij}} - \frac{A}{2} & t \le \tau_f \\ \overline{s_{ij}} + \frac{A}{2} - \frac{A(t - \tau_f)}{\tau_s} & \tau_f < t \le \tau_s \end{cases}, \tag{27}$$

where $A = s_{ij}^{\text{clamped}} - s_{ij}^{\text{free}}$ and $\overline{s_{ij}} = \frac{1}{2}(s_{ij}^{\text{clamped}} + s_{ij}^{\text{free}})$.

The nonlinearity $g$ is 0 below a threshold magnitude $\theta_g$ and linear elsewhere:

$$g(u) = \begin{cases} u & |u| \ge \theta_g \\ 0 & |u| < \theta_g \end{cases}. \tag{28}$$

The kernel $K$ varies in exact functional form, but has a characteristic timescale $\tau_K$.

We show in Section S1 that our weight update Eq. (6) approximates an ideal contrastive rule:

$$\Delta w_{ij} \approx \epsilon \left(s_{ij}^{\text{clamped}} - s_{ij}^{\text{free}}\right), \tag{29}$$

if the following conditions are met:
1. $I = \int_0^\infty K(t) \, dt = 0$;
2. $\tau_K \ll \tau_f \ll \tau_s$.

We specify more exact numerical details of weight update computations in Section S9.

### Training neural networks with Temporal Contrastive Learning
We use our Temporal Contrastive Learning framework in a neural network model for MNIST recognition. Details of the network architecture are available in Section S4. Each node of the network is described by a state $x$. During inference, the dynamics of the network minimizes:

$$E(x) = \frac{1}{2}\Sigma_i x_i^2 - \frac{1}{2}\Sigma_{i,j} w_{ij}\eta_i\eta_j - \Sigma_i b_i\eta_i, \tag{30}$$

where $b_i$ is the bias of node $i$, $w_{ij}$ is the weight of the synapse connecting nodes $i$ and $j$, and the indices $i, j$ run over the node indices of all layers (input, hidden, and output). $\eta(x)$ is an activation function clip$(x, 0, 1)$. During training of the network, the network minimizes:

$$F(x; t) = E(x) + \frac{\beta(t)}{2}\Sigma_{o=0}^9 (x_o - \upsilon_o^{\text{label}})^2. \tag{31}$$

Here, $\upsilon_o^{\text{label}}$ is the one-hot encoding of the desired MNIST output class, and the $x_o$ are the states of the 10 neurons in the output layer. $\beta$ is a clamping parameter which varies in time as a sawtooth function.

As $\beta$ varies, the synaptic current $s_{ij} = \eta(x_i)\eta(x_j)$ changes as the network minimizes energy. These changes inform the updates of weights $w_{ij}$ according to Eq. (6), with a layer-dependent nonlinearity $g$ and a sinusoidal kernel $K$. Further details of training and inference protocols are provided in Section S4.

## Data availability
The data generated in this study may be found in the following figshare database: https://doi.org/10.6084/m9.figshare.25057793.

## Code availability
Code for training neural networks to perform MNIST recognition can be found at https://github.com/falkma/ContrastiveMemory-Exp. Code for understanding tradeoffs between protocol time and dissipation can be found at https://github.com/atstrupp/ContrastiveMemory-SynapseAnalysis.

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

## Acknowledgements

The authors thank Lauren Altman, Marjolein Dijkstra, Sam Dillavou, Douglas Durian, Andrea Liu, Marc Miskin, Krishna Shrinivas, Menachem Stern, and Erik Winfree for discussion. AM acknowledges support from the NSF through DMR-2239801 and by NIGMS of the NIH under award number R35GM151211. MJF is supported by the Eric and Wendy Schmidt AI in Science Postdoctoral Fellowship, a Schmidt Sciences program. This work was supported by the NSF Center for Living Systems (NSF grant no. 2317138) and by the University of Chicago's Research Computing Center. This work was performed in part at the Aspen Center for Physics, which is supported by National Science Foundation grant PHY-2210452.

## Author contributions

MJF, AM, ATS conceived the study, developed methodology, and wrote the original draft. MJF, ATS developed software and performed investigation. MJF, BS, ATS provided formal analysis. MJF, AM supervised and acquired funding. All authors contributed to review and editing.

## Competing interests

The authors declare no competing interests.
