## [Transparent Peer Review file · Nature Communications]

Temporal Contrastive Learning through implicit non-equilibrium memory

Corresponding Author: Professor Arvind Murugan

Version 0:

Reviewer comments:

Reviewer #1

(Remarks to the Author)
Dear Editor,

Falk et al. have demonstrated a new algorithmic technique that is essentially an implicit learning approach within the broad class of energy-based contrastive learning. This approach reduces the memory required to implement energy-based contrastive algorithms. The paper is mostly a mathematical exercise. The authors finally argue that many physical systems in nature naturally implement such implicit learning techniques. The main claim is that such techniques can replace backpropagation (the most widely used algorithm today), and thereby reduce the energy and cost of AI training.

The technique is an interesting mathematical primitive, but has fundamental limitations. Here are my main comments.

1. Memory is not the main issue with energy-based algorithms. The main reason such algorithms cannot compete with backpropagation is that they need many cycles to settle/relax to some minimum/equilibrium/steady-state. In fact, such a settling process scales very poorly with size. Finding the energy minimum is effectively identifying the eigenvalues of a network, which requires a number of steps, which scales as a power of the network/matrix size (see for example: Adiabatic quantum computation. Reviews of Modern Physics, 90, 015002, 2018). Without addressing such a pressing issue, saying that this effort improves memory footprint of energy-based algorithms is not a compelling story and is likely not going to be of interest to a broad audience. Furthermore, there are other issues with these algorithms, such as the need for symmetry, which prevent them from competing with backpropagation. The authors rightly note that "However, a wide class of 'contrastive learning' algorithms promise better results at the cost of complexity in training". However, the authors don't list the real complexities, and they don't address such complexities either.

If the authors want to really claim that they are doing something better than backpropagation, they need to show scalability and maintaining of accuracy of training larger models. Specifically, they need to show that the energy/number of steps in this algorithm is clearly better than backpropagation.

2. The proposed algorithm is not implemented in a comparable way to backpropagation. They seem to have trained a small-scale network on MNIST (a toy-sized dataset), but there are no comparisons to other algorithms (esp. backprop). I would expect a fair comparison on many metrics (e.g., number of steps, energy, algorithmic complexity class, $O(n)$, etc.), particularly when training larger networks/models.

3. The authors argue that many physical and naturally occurring systems may use the kind of implicit training designed by the authors. It is not clear if they are saying that their algorithm explains many naturally occurring systems. If so, a reader would want to know how it is beneficial to neural network training on hardware (goes back to comments (1) and (2)). Also, the authors have cited several of their past work using some form of implicit learning (e.g., Stern et al.). Is it a conclusion that this algorithm explains the learning technique in many such past works? Or, are there examples of past works where this algorithm explains the learning technique?

4. The authors imply that there may be a need for new hardware to support this algorithm. It is unclear if this algorithm can be implemented on any digital hardware and still outperform other algorithms, or if it needs a new type of hardware (such as analog) to be successful in terms of performance. At the least, your readers would expect a rigorous simulation.

(Remarks on code availability)

Reviewer #2

(Remarks to the Author)

The manuscript entitled "Contrastive learning through implicit non-equilibrium memory" shows how integral feedback dynamics at individual neural networks allow for contrastive learning without the need for explicit memory as required for comparing free and clamped. The workings of the mechanism is derived analytically, established in its power numerically before numerically investigating and analytically deriving scaling relationships regarding its speed-accuracy tradeoff and dissipation cost. Last the authors discuss how their integral feedback could be active in diverse systems featuring Hebbian-like learning. Their work strikes me as not only putting forward a novel concept of huge advancement in contrastive learning but in fact by a beautiful interplay of numerical investigation and theoretical, analytical mechanistic insight. I only have a few stylistic revisions I request before publication.

Minor comments:

Lines 38: the concept of 'energy-based rules' is not introduced at this point

Lines 38-45: the content seems somehow doubled in these two sentences. It would help the accessibility of the manuscript to give this entire paragraph a good polish.

Appendix A and Model introduction are contradictory regarding the choice of the kernel function K . While the Appendix (lines 1065/1066) states that K is assumed to be an arbitrary function with characteristic timescale τ_K the model introduction in the manuscript (lines 192/193) refers to an exponential decay times a polynomial. Reading through the model introduction I was very frustrated by not being given the polynomial form and would recommend that the author provide the stronger general result shown in the appendix in the main manuscript and only refer to the exponential decay times polynomial as an example. Also moving the reasoning of lines 1052 to 1081 to the main as key mechanistic insight is highly recommended. If need to shorten it would for me be in the dissipation part which reads rather technical and almost like a second story.

(Remarks on code availability)

Reviewer #3

(Remarks to the Author)

In this paper, Falk et al. investigate implementations of CL that replace such two-point time-comparisons by a temporal convolution with a non-monotonic memory kernel, followed by a threshold nonlinearity, that are arguably more amenable to physical implementation. The authors identify a hierarchy of timescales such that, when the system is driven fast towards the clamped and then slowly decays back to a free state, their proposed process yields an accurate approximation to the CL update. These conditions are supplemented with a rich analysis of the memory kernel, which includes a link between its integral and the energy costs of CL in physical systems. Finally, the paper suggests that the memory kernel convolution could be implemented with an integral feedback controller, a creative proposal that allows establishing high-level connections to a host of different physical systems, that could be compatible with CL.

This is a technically solid paper, written for the most part in an accessible, clear and enticing way. The analyses are insightful and elegant. I thought that the connection to integral feedback control, using its well-known overshoot failure mode, was a highly creative idea. Given the interdisciplinary nature of this research, the accessible writing, and the interesting findings, I think that this paper is worthy of publishing in a broad-audience journal such as Nature Communications.

I do have however a few questions that I would like to see addressed by the authors before I consider the paper ready for publication.

- For most of the systems discussed in Fig. 7 and the related main text sections, it wasn't clear to me what the exact intended claims are. Do the authors wish to suggest that certain systems might be easily modifiable so as to implement CL (e.g., molecular systems)? Do they wish to speculate that some already do (e.g., the brain)?

- My understanding is that for some of the systems the authors actually have some quite concrete ideas in mind. I think that the paper would be greatly improved if these were spelled out in more detail, for example in the appendix. I found the current level of explanation too high to follow. It's of course fine if the ideas are more speculative in some cases; it would, however, be good to state this explicitly.

- Regarding biological neural networks, as the authors comment, there is a loose high-level similarity to STDP. In the same vein as my previous comment, it would be great if the authors could speculate on which synaptic mechanisms could implement the required synaptic feedback controller; with some luck, prior research on STDP, and synaptic plasticity more generally, already has some hints the authors might take.

- The paper discusses the original two-phase CL update at length. However, multiple previous studies have proposed alternatives that are more compatible with physical systems. Two past proposals appear to be of particular relevance. First, the equilibrium propagation paper by one of the co-authors [30] suggests to integrate differences on the path from the free to

the clamped state (see also Ref. [a]), which is conceptually not so far from what the authors propose here. Second, Baldi and Pineda study oscillatory teaching signals that make the system alternate between free and clamped states, as synapses continuously integrate phase-modulated synaptic currents. The relative merits of the new approach as well as its relationship to these older ideas should be discussed in depth.

[a] Equilibrium Propagation with Continual Weight Updates. Maxence Ernout, Julie Grollier, Damien Querlioz, Yoshua Bengio, Benjamin Scellier, arXiv, 2020.

(Remarks on code availability)

Reviewer #4

(Remarks to the Author)

The manuscript "Contrastive Learning through implicit non-equilibrium memory" proposes a novel way to implement a contractive divergence strategy to train quadratic energy functions without the need to store the "clamped" and the "free" configurations in memory. While the proposal may be interesting, I find the current version of the paper extremely difficult to read for non-specialists in the field of biological networks and difficult to apply to alternative problems in practice, so I cannot recommend its publication in Nature Communications.

In particular, it is not clearly defined what the authors mean by "clamped" and "free" states, nor is the procedure by which they achieve these states clearly defined. In contrastive divergence, the positive and negative terms are usually averages of the correlations in the data set and in the equilibrium distribution of the model. I assume that the authors mean here the instantaneous configurations, i.e. a Hebbian learning procedure in combination with a dreaming strategy that is not explained. This might not be the case, because the way both types of states are reached is not explained, not even in the numerical test discussed.

I would like the authors to explain why it is much easier to switch between the two states and store the integral of the convolution with one kernel than to store two configurations. How do the neurons know how to switch back and forth between the two states? If the alternation is not complicated, why not simply store the difference between the two?

The numerical test using this training scheme to predict the labels from MNIST is incomprehensible to me. How do you go from a pairwise model to predicting 10 labels? The authors should explain the set-up and how this machine was trained.

Can their networks be trained to work as generative models or at least store patterns?

Can this memory method be related to the recently proposed out-of-equilibrium recipe for training energy-based models in general? ("Explaining the effects of non-convergent sampling in the training of Energy-Based Models" ICML 2023)

(Remarks on code availability)

Reviewer #5

(Remarks to the Author)

Dear Editor and Authors,

This is an interesting paper. However, in my view, there is significant margin for improvement. Please find my feedback below.

Most importantly, the proposed approach has very strong links to STDP, but the authors only mention STDP once and very briefly, only to assert that the manuscript's approach is significantly different. This would need a rigorous substantiation. I must say that I doubt that an attempt to do this will result in a truly significant difference. Specifically, it appears that the mechanism described in this paper is a rate-based description of STDP's outcome, if the authors' s_{ij} is considered as a rate. There are multiple prior works that show the relationship between spike-based and rate-based learning. These works include rate-based interpretations of STDP (e.g. [r1]), and also the reverse, i.e. derivations of optimal STDP kernels from rate-encoded inputs (e.g. Section S6.1 in [r2]).

This latter type of prior work also relates to the section of the present manuscript that examines the kernel shape or area. That section also relates to a further large body of literature that examines the effect of STDP kernel shapes on learning but is uncited.

Further aspects of the manuscript that I recommend improving are the following:

- Equilibrium Propagation is used and modified, but it and its specific modifications are not described.
- The aspects of the manuscript that focus on "dissipation" seem interesting, but dissipation as a concept is not introduced. For example, what is dissipated? What is meant by "dissipative integral feedback dynamics"?
- The manuscript mentions that Hebbian and anti-Hebbian plasticity are commonly applied in separate phases, but that does not acknowledge previous work that has indeed applied them in a single phase and showed that this works relatively well (e.g. [r3, r4]). The present work seems to also use a single learning mode at the synapse level, but it does switch modes by varying the clamping signal, and by expanding an individual training example from a single timestep to an extended duration. This is not commented on. What is the advantage compared to these other works?

- Even compared to Equilibrium Propagation, and the other algorithms that do involve two phases, could the authors increase the clarity on the benefit of the present mechanism? The key seems to be in the contrast between "implicit" and "explicit" memory, in the authors' terms. Can this be clarified further?

- Experiments are very limited. Only a one-hidden-layer network is used, only MNIST classification is tested, and there are no comparisons with alternative learning rules.

- The synapses are bidirectional, which is not plausible biologically.

More generally, it would be very helpful if the authors' response could also include a statement on the broader significance and impact of this manuscript.

Sincerely,
Timos Moraitis

References

[r1] Burkitt, Anthony N., Hamish Meffin, and David B. Grayden. "Spike-timing-dependent plasticity: the relationship to rate-based learning for models with weight dynamics determined by a stable fixed point." *Neural Computation* 16.5 (2004): 885-940.

[r2] Moraitis, Timoleon, Abu Sebastian, and Evangelos Eleftheriou. "Optimality of short-term synaptic plasticity in modelling certain dynamic environments." *arXiv preprint arXiv:2009.06808* (2020).

[r3] Krotov, Dmitry, and John J. Hopfield. "Unsupervised learning by competing hidden units." *Proceedings of the National Academy of Sciences* 116.16 (2019): 7723-7731.

[r4] Journé, Adrien, Hector Garcia Rodriguez, Qinghai Guo, and Timoleon Moraitis, "Hebbian Deep Learning Without Feedback." *The Eleventh International Conference on Learning Representations (ICLR 2023)*.

(Remarks on code availability)
I could not find any code attached.

Version 1:

Reviewer comments:

Reviewer #1

(Remarks to the Author)
Dear Editor,

The authors have clarified some of the issues I raised in the first round of reviews. While their responses made the scope of the manuscript clear, they also raise major questions that need to be resolved. Most importantly, the authors claim that they are not developing an algorithm that can compete with the dominant backpropagation (that is usually run on digital hardware), but they are focused on optimizing an algorithm for non-digital (mostly analog hardware). To that end, the authors need to provide clear and fair comparisons among the main existing non-backpropagation algorithms. Please compare the performance, time/cycles taken, energy consumed (using simple assumptions on analog hardware), etc. You could use a small data set to demonstrate your performance metrics. This exercise need not be very laborious - it can be on toy-sized models and datasets. But this exercise is important in a paper published in a general-audience journal, without which it becomes a mere mathematical demonstration.

(Remarks on code availability)

Reviewer #2

(Remarks to the Author)
The authors incorporated all my suggestions. I recommend publication.

(Remarks on code availability)

Reviewer #3

(Remarks to the Author)
I thank the authors for the changes made to the manuscript, and for replying to my questions.

(Remarks on code availability)

Version 2:

Reviewer comments:

Reviewer #1

(Remarks to the Author)

-

(Remarks on code availability)

REVIEWER COMMENTS

Reviewer #1 (Remarks to the Author):

Dear Editor,

Falk et al. have demonstrated a new algorithmic technique that is essentially an implicit learning approach within the broad class of energy-based contrastive learning. This approach reduces the memory required to implement energy-based contrastive algorithms. The paper is mostly a mathematical exercise. The authors finally argue that many physical systems in nature naturally implement such implicit learning techniques. The main claim is that such techniques can replace backpropagation (the most widely used algorithm today), and thereby reduce the energy and cost of AI training.

We thank the reviewer for the detailed review; however, the motivation for our work is a bit different and also broader than implied here. We have edited our introduction to be clear about the motivation that is **not** to create an algorithm to replace backpropagation on digital hardware but rather a learning algorithm on hardware (both natural and neuromorphic) with no central processor is available and hence backprop is not available (Introduction, line 73):

Here, our primary contribution is to show how a ubiquitous process - integral feedback control - can allow for contrastive learning without the complexities of explicit memory of free and clamped states or switching between Hebbian and anti-Hebbian update modes. Further, in our method, computing the contrastive weight update signal and then performing the update of weights are not separate steps involving different kinds of hardware but are the same unified *in situ* operation. This approach to contrastive learning, which we call 'Temporal Contrastive Learning through feedback control', allows a wide range of physical and biological platforms without central processors to 'physically learn' (i.e., autonomously learn) novel functions through contrastive learning methods.

In other words, physical processes that naturally implement this learning technique are not an afterthought but a primary motivation for this work. In these contexts, no central processor is available and thus backpropagation is not an option. These systems include the many natural systems described in the paper (the brain, molecular systems in the cell, biomechanics, flow networks) as well as some neuromorphic platforms that we have now extensively cited as detailed below.

We respond point by point to the following comments:

The technique is an interesting mathematical primitive, but has fundamental limitations. Here are my main comments.

We have reordered the reviewer's comments below to move point (4) up front since the answer to that critical question helps set context for the response to the other comments.

4. The authors imply that there may be a need for new hardware to support this algorithm. It is unclear if this algorithm can be implemented on any digital hardware and still outperform other algorithms, or if it needs a new type of hardware (such as analog) to be successful in terms of performance. At the least, your readers would expect a rigorous simulation.

Indeed, we have edited our paper to explicitly clarify, right from the beginning, that our method is not for digital hardware but rather for contexts where a central processor (and backprop) is NOT available. In particular, we have:

1. Rewritten the introduction to highlight interest in strongly-constrained computational substrates. The new introduction emphasizes that our motivation is a learning algorithm in strongly-constrained physical, biological and neuromorphic systems where a central processor (and backpropagation) is not available.
 - a. For example, line 73: "Here, our primary contribution is to show how a ubiquitous process - integral feedback control - can allow for contrastive learning without the complexities of explicit memory of free and clamped states or switching between Hebbian and anti-Hebbian update modes. ... This approach to contrastive learning, which we call 'Temporal Contrastive Learning through feedback control', allows a wide range of physical and biological platforms without central processors to 'physically learn' (i.e., autonomously learn) novel functions through contrastive learning methods."
2. Expanded the level of detail in making suggestions for implementation in physical systems (e.g., giving explicit equations for synaptic unit dynamics).
 - a. For example, for mechanical systems, line 652: "We add a simple twist to this Hebbian framework in order to make such systems capable of contrastive learning; we add a viscoelastic element with a faster timescale of relaxation compared to the timescale of bond softening. In reduced-order modeling of viscoelastic materials, a common motif is that of a viscous dashpot connected to an elastic spring in series, also known as a Maxwell material unit. Such units obey the following simple equation:

$$\frac{\dot{u}}{k} + \frac{u}{\gamma} = \dot{s}$$

where u is the stress in the unit, x is the strain in the unit, k is the Hookean modulus of the spring and γ is the Newtonian viscosity." We have provided analogous guidance for mechanochemical, nano-fluidic, and chemical reaction-based systems.

- b. This expansion includes both natural (systems described in Fig. 7b,c) and engineered systems (systems described in Fig. 7d,e).

3. Compared our approach in more detail to other algorithms meant for such constrained analog hardware (e.g., Equilibrium Propagation implemented with memory units).
 - a. For example, in a new “Related Work Summary” section, line 254: “Several works have proposed ways in which contrastive learning can be generated in natural and engineered systems, such as: utilizing explicit memory[27, 28]; having two globally coordinated phases each with low learning rates[46]; having two copies of the system[47-49]; having two physically distinct kinds of signals[15]; or using continually-running oscillations in the learning rules[16, 24]. All these methods require globally switching the system between two phases or exploit hardware-specific mechanisms. The closest to the TCL approach here is ‘continual EP’[50], but it also requires globally switching between phases of learning.”
 - b. The “Related Work Summary” section serves as a pointer to a newly-written, more in-depth “Related Work” Appendix B.

4. As suggested by the reviewer, we also compare the time costs of this method to those of contrastive methods which utilize explicit memory storage; we make this explicit now in the Discussion (line 734):
 - a. Our framework has several limitations quantified by speed-accuracy and speed-energy tradeoffs derived here, in addition to the general concerns about adiabatic protocols in contrastive methods[25, 60]. The time costs in our scheme (e.g. those shown in Fig. 4) are inherently larger compared to other Equilibrium Propagation-like methods which use explicit memory and global signals to switch between wake and sleep, thereby avoiding slow ramping protocols.

1. Memory is not the main issue with energy-based algorithms. The main reason such algorithms cannot compete with backpropagation is that they need many cycles to settle/relax to some minimum/equilibrium/steady-state. In fact, such a settling process scales very poorly with size. Finding the energy minimum is effectively identifying the eigenvalues of a network, which requires a number of steps, which scales as a power of the network/matrix size (see for example: Adiabatic quantum computation. Reviews of Modern Physics, 90, 015002, 2018). Without addressing such a pressing issue, saying that this effort improves memory footprint of energy-based algorithms is not a compelling story and is likely not going to be of interest to a broad audience. Furthermore, there are other issues with these algorithms, such as the need for symmetry, which prevent them from competing with backpropagation. The authors rightly note that "However, a wide class of 'contrastive learning' algorithms promise better results at the cost of complexity in training". However, the authors don't list the real complexities, and they don't address such complexities either.

If the authors want to really claim that they are doing something better than backpropagation, they need to show scalability and maintaining of accuracy of training

larger models. Specifically, they need to show that the energy/number of steps in this algorithm is clearly better than backpropagation.

As stated above, we do not “want to [...] claim that [we] are doing something better than backpropagation,”; our ideas are meant for analog hardware (both natural and engineered neuromorphic platforms) in which a central processor (like a GPU) is not available.

As we state in the revised manuscript, line 35:

Excitingly, local learning rules also open the possibility of endowing computationally-constrained physical systems with functionality through an in situ period of training, rather than by prior backpropagation-aided design on a computer[5-21]. In these settings, sometimes as simple as chemical reactions within a cell or a mechanical material, there is no centralized control that would allow backpropagation to be a viable method of learning.

Additionally, we have now edited the manuscript to be more accurate about the shortcomings of this approach. As pointed out by the reviewer, the primary cost of this approach is time. We now cite the review the reviewer suggested, e.g. when we describe the application of our method (line 332):

We assume that both these timescales are quasi-static compared to any system-internal energy relaxation timescales[60]. This restriction to adiabatic protocols is a potential limitation to contrastive methods in general; details depend on the system and some works present viable workarounds[25, 61].

We additionally discuss physical relaxation time effects in our expanded discussion of system examples in Fig. 7 (line 671):

Note that, in contrast to the solution from idealized Eqs.17, 18, the material moduli k_{ij} are involved both as learning degrees of freedom and as elements of the integral feedback control.

See lines 663 and 698 for expressions of relaxation times in physical systems which compete with protocol timescales.

But as above, we emphasize that our goal is **not** a computer algorithm that is “better than backpropagation” on digital hardware; our focus is on analog hardware contexts (both natural like molecular networks in a cell or synthetic like some neuromorphic platforms) where backpropagation is not available. In these systems, the benefit is not to energy savings compared to backpropagation; the benefit is simply that training can be performed at all.

2. The proposed algorithm is not implemented in a comparable way to backpropagation. They seem to have trained a small-scale network on MNIST (a toy-sized dataset), but

there are no comparisons to other algorithms (esp. backprop). I would expect a fair comparison on many metrics (e.g., number of steps, energy, algorithmic complexity class, $O(n)$, etc.), particularly when training larger networks/models.

As stated above, our goal is not to develop an algorithm meant to run on digital hardware; there is no way to implement backprop on the family of hardware discussed in this paper because such hardware lacks a central processor.

The relevant comparisons are to other distributed algorithms that could work on similarly constrained analog hardware; the revised manuscript makes this clear and states, e.g., the extra time cost of our algorithm compared to some of these distributed/local algorithms (Discussion, line 734):

Our framework has several limitations quantified by speed-accuracy and speed-energy tradeoffs derived here, in addition to the general concerns about adiabatic protocols in contrastive methods[25, 60]. The time costs in our scheme (e.g. those shown in Fig. 4) are inherently larger compared to other Equilibrium Propagation-like methods which use explicit memory and global signals to switch between wake and sleep, thereby avoiding slow ramping protocols.

These comparisons are also discussed in more detail in our new “Related Works Summary” section in the main text and “Related Works” appendix.

As for backpropagation, our motivation is to ask if natural processes can be a training algorithm on hardware where backpropagation is not possible. The comparisons asked for would make sense if we were proposing a digital algorithm that needs to be compared to another digital algorithm on similar hardware. But while a range of natural and engineered systems here can potentially learn by exploiting integral feedback, they cannot do backpropagation autonomously.

3. The authors argue that many physical and naturally occurring systems may use the kind of implicit training designed by the authors. It is not clear if they are saying that their algorithm explains many naturally occurring systems. If so, a reader would want to know how it is beneficial to neural network training on hardware (goes back to comments (1) and (2)). Also, the authors have cited several of their past work using some form of implicit learning (e.g., Stern et al.). Is it a conclusion that this algorithm explains the learning technique in many such past works? Or, are there examples of past works where this algorithm explains the learning technique?

Indeed, this aspect raised by the reviewer is not an afterthought but one of the primary motivations for this work.

As our revised manuscript now explains, our focus is to show how many natural (physical + biological) systems can exploit learning processes; our work also opens doors in synthetic

biology and materials science for trainable materials and circuits. For example, in the Introduction, line 35:

Excitingly, local learning rules also open the possibility of endowing computationally-constrained physical systems with functionality through an in situ period of training, rather than by prior backpropagation-aided design on a computer[5-21]. In these settings, sometimes as simple as chemical reactions within a cell or a mechanical material, there is no centralized control that would allow backpropagation to be a viable method of learning.

And line 73:

Here, our primary contribution is to show how a ubiquitous process - integral feedback control - can allow for contrastive learning without the complexities of explicit memory of free and clamped states or switching between Hebbian and anti-Hebbian update modes. ... This approach to contrastive learning, which we call 'Temporal Contrastive Learning through feedback control', allows a wide range of physical and biological platforms without central processors to 'physically learn' (i.e., autonomously learn) novel functions through contrastive learning methods.

Our works reviews and cites a large body of literature describing a diverse set of biological and physical hardware which are autonomously trained to perform computation, e.g. mechanical networks[1-4], flow networks[5-7], regulatory networks[8, 9], and chemical reaction networks[10, 11], non-exhaustively. Our work provides a novel way to introduce a contrastive learning framework into these computation-capable but constrained systems. The equations presented in the newly expanded "Realizing memory kernels through integral feedback" section provide quantitative guidance on how to do so.

[1] Pashine, Nidhi, et al. "Directed aging, memory, and nature's greed." *Science advances* 5.12 (2019): eaax4215.

[2] Rocks, Jason W., et al. "Designing allostery-inspired response in mechanical networks." *Proceedings of the National Academy of Sciences* 114.10 (2017): 2520-2525.

[3] Stern, Menachem, et al. "Supervised learning in physical networks: From machine learning to learning machines." *Physical Review X* 11.2 (2021): 021045.

[4] Anisetti, Vidyesh Rao, et al. "Frequency propagation: Multimechanism learning in nonlinear physical networks." *Neural Computation* 36.4 (2024): 596-620.

[5] Kramar, Mirna, and Karen Alim. "Encoding memory in tube diameter hierarchy of living flow network." *Proceedings of the national academy of sciences* 118.10 (2021): e2007815118.

[6] Marbach, Sophie, et al. "Vein fate determined by flow-based but time-delayed integration of network architecture." *Elife* 12 (2023): e78100.

[7] Anisetti, Vidyesh Rao, Benjamin Scellier, and Jennifer M. Schwarz. "Learning by non-interfering feedback chemical signaling in physical networks." *Physical Review Research* 5.2 (2023): 023024.

- [8] Zhu, Ronghui, et al. "Synthetic multistability in mammalian cells." *Science* 375.6578 (2022): eabg9765.
- [9] Parres-Gold, Jacob, et al. "Principles of Computation by Competitive Protein Dimerization Networks." *bioRxiv* (2023).
- [10] Poole, William, et al. "Chemical boltzmann machines." *DNA Computing and Molecular Programming: 23rd International Conference, DNA 23, Austin, TX, USA, September 24–28, 2017, Proceedings 23*. Springer International Publishing, 2017.
- [11] Poole, William, et al. "Detailed balanced chemical reaction networks as generalized Boltzmann machines." *arXiv preprint arXiv:2205.06313* (2022).

Reviewer #2 (Remarks to the Author):

The manuscript entitled “Contrastive learning through implicit non-equilibrium memory” shows how integral feedback dynamics at individual neural networks allow for contrastive learning without the need for explicit memory as required for comparing free and clamped. The workings of the mechanism is derived analytically, established in its power numerically before numerically investigating and analytically deriving scaling relationships regarding its speed-accuracy tradeoff and dissipation cost. Last the authors discuss how their integral feedback could be active in diverse systems featuring Hebbian-like learning. Their work strikes me as not only putting forward a novel concept of huge advancement in contrastive learning but in fact by a beautiful interplay of numerical investigation and theoretical, analytical mechanistic insight. I only have a few stylistic revisions I request before publication.

Minor comments:

1. Lines 38: the concept of ‘energy-based rules’ is not introduced at this point

We have now removed the premature reference to energy-based rules.

2. Lines 38-45: the content seems somehow doubled in these two sentences. It would help the accessibility of the manuscript to give this entire paragraph a good polish.

We have removed the first sentence (which previously began on line 38). Our streamlined and more polished paragraph now reads:

In particular, a large class of local ‘contrastive learning’ algorithms (contrastive Hebbian learning[22-24], Contrastive Divergence[25], Equilibrium Propagation[26]) promise impressive results, but make requirements on the capabilities of a single synapse (or more generally, on learning degrees of freedom). While details differ, training weights generally are updated based on the difference between Hebbian-like rules applied during a ‘clamped’ state that roughly corresponds to desired behaviors and a ‘free’ state that corresponds to the spontaneous (and initially undesirable) behaviors of the system.

3. Appendix A and Model introduction are contradictory regarding the choice of the kernel function K . While the Appendix (lines 1065/1066) states that K is assumed to be an arbitrary function with characteristic timescale τ_K the model introduction in the manuscript (lines 192/193) refers to an exponential decay times a polynomial. Reading through the model introduction I was very frustrated by not being given the polynomial form and would recommend that the author provide the stronger general result shown in the appendix in the main manuscript and only refer to the exponential decay times polynomial as an example.

We have now clarified in line 194 that the exponential decay-polynomial product kernel is just an example:

While most physical and biological systems have some form of memory, the most common memory is described by monotonic kernels, for example $K(t-t') \sim e^{-(t-t')/\tau_K}$. But a broad class of non-equilibrium systems exhibit memory with a non-monotonic kernel with both positive and negative lobes, for example $K(t-t') \sim e^{-(t-t')/\tau_K} f(t-t')$, with $f(t-t')$ a polynomial[31,38-45].

Furthermore, following the reviewer's advice, we have left the subsequent explanation of the intuition of our method at the level of a general nonmonotonic kernel with timescale τ_K , as opposed to specifying a particular example.

4. Also moving the reasoning of lines 1052 to 1081 to the main as key mechanistic insight is highly recommended. If need to shorten it would for me be in the dissipation part which reads rather technical and almost like a second story.

We have now streamlined the main text portion describing our method, in particular in describing our temporal protocols (lines 221):

The only time-dependence comes from training examples being presented in a time-dependent way. For simplicity, we consider a sawtooth-like training protocol, where input and output neurons $\eta_i(t)$ alternate between free and clamped states over time (Fig. 2C). In this sawtooth protocol, the free-to-clamped change is fast (time τ_f) while the clamped-to-free relaxation is slow (time $\tau_s > \tau_f$); see Appendix A for further detail.

This has allowed us to move important mechanistic insight from the section the reviewer references into the main text (line 236):

Intuitively, the kernel K computes the approximate (finite) time derivative of $s_{ij}(t)$; this is what requires the kernel timescale $\tau_K \ll \tau_f$. The rapid rise of the sawtooth from free to clamped states results in a large derivative that exceeds the threshold θ_g in $g(x)$ and provides the necessary contrastive learning update. The slow relaxation from clamped to free has a small time-derivative that is below θ_g . The ability to distinguish free-to-clamped versus clamped-to-free transitions is what requires $\tau_f \ll \tau_s$.

Reviewer #3 (Remarks to the Author):

In this paper, Falk et al. investigate implementations of CL that replace such two-point time-comparisons by a temporal convolution with a non-monotonic memory kernel, followed by a threshold nonlinearity, that are arguably more amenable to physical implementation. The authors identify a hierarchy of timescales such that, when the system is driven fast towards the clamped and then slowly decays back to a free state, their proposed process yields an accurate approximation to the CL update. These conditions are supplemented with a rich analysis of the memory kernel, which includes a link between its integral and the energy costs of CL in physical systems. Finally, the paper suggests that the memory kernel convolution could be implemented with an integral feedback controller, a creative proposal that allows establishing high-level connections to a host of different physical systems, that could be compatible with CL.

This is a technically solid paper, written for the most part in an accessible, clear and enticing way. The analyses are insightful and elegant. I thought that the connection to integral feedback control, using its well-known overshoot failure mode, was a highly creative idea. Given the interdisciplinary nature of this research, the accessible writing, and the interesting findings, I think that this paper is worthy of publishing in a broad-audience journal such as Nature Communications.

I do have however a few questions that I would like to see addressed by the authors before I consider the paper ready for publication.

1. For most of the systems discussed in Fig. 7 and the related main text sections, it wasn't clear to me what the exact intended claims are. Do the authors wish to suggest that certain systems might be easily modifiable so as to implement CL (e.g., molecular systems)? Do they wish to speculate that some already do (e.g., the brain)?

We thank the reviewer for pushing us to clarify the specificity and intent of Fig. 7. As the reviewer says, we mean to suggest that certain systems would be easily manipulated to perform CL. We now state in the text (line 526):

We have established that contrastive learning can arise naturally as a consequence of non-equilibrium memory as captured by a non-monotonic memory kernel K . Here, we argue that the needed memory kernels K in turn arise naturally as a consequence of integral feedback control in a wide class of physical systems. Hence many simple physical and biological systems can be easily modified and manipulated to undergo contrastive learning.

2. My understanding is that for some of the systems the authors actually have some quite concrete ideas in mind. I think that the paper would be greatly improved if these were spelled out in more detail, for example in the appendix. I found the current level of

explanation too high to follow. It's of course fine if the ideas are more speculative in some cases; it would, however, be good to state this explicitly.

We thank the reviewer for encouraging us to describe more explicitly the systems we have in mind that are potential substrates for our Temporal Contrastive Learning approach. Indeed, we have quite concrete ideas; we have now included substantial new text and added Eqs. 19-25, which formally describe how a diverse array of chemical, electrical, and mechanical materials can host integral feedback control and implement contrastive learning rules.

3. Regarding biological neural networks, as the authors comment, there is a loose high-level similarity to STDP. In the same vein as my previous comment, it would be great if the authors could speculate on which synaptic mechanisms could implement the required synaptic feedback controller; with some luck, prior research on STDP, and synaptic plasticity more generally, already has some hints the authors might take.

There is a rich literature which proposes molecular/cellular mechanisms for STDP curves such as those reported in Bi and Poo (2000). However, we think attempting to find such connections might prove confusing to readers, as we feel our method and STDP are fundamentally different, despite their high level similarity (overview on line 268). To this end, we have provided a substantial amount of new text in a new Related Works Appendix B; please see response to Reviewer #5 as well:

f. Connection with STDP. Spike-timing-dependent plasticity (STDP) (Bi and Poo [51]) can be formulated as a learning rule for asymmetric networks where a synapse distinguishes between the time of presynaptic spikes t_k and the time of postsynaptic spikes t_j :

$$\Delta w_{ij} = f(t_j - t_k) \quad (\text{B8})$$

where f has a positive lobe for positive arguments and a negative lobe for negative arguments, and approximately vanishes outside a pairing range $[-\tau, \tau]$.

A large body of work has interrogated the equivalence of spike- and rate-based formulations of STDP [54, 55]. In particular, Xie and Seung [42] considered additive STDP updates for δ -function timeseries $s_j(t)$, written as:

$$\Delta w_{ij} = \int_0^T dt_k \int_{-\infty}^{\infty} dt_j f(t_j - t_k) s_j(t_j) s_k(t_k) \quad (\text{B9})$$

over a time period $[0, T]$. They showed that Eq. B9 can be approximated as rate-based formulations depending on firing rates $\nu_j(t)$ when rates of spiking vary slowly compared to the pairing range:

$$\Delta w_{ij} = \int_0^T dt [\beta_0 \nu_j(t) + \beta_1 \dot{\nu}_j(t)] \nu_k(t) \quad (\text{B10})$$

where β_0 is the integral of f over the pairing range, and β_1 is the first moment of f over the pairing range.

In the case where $\beta_0 = 0$ and $\beta_1 > 0$, Eq. B10 can be interpreted as the asymmetric version of the nudge phase update of Continual EP (Eq. B6):

$$\frac{dw_{ij}}{dt} = \beta_1 \dot{\nu}_j(t) \nu_k(t). \quad (\text{B11})$$

Martin *et al.* [53] proposed a spiking version of contrastive learning to train spiking networks with bidirectional weights. Similar to the Continual EP approach, in the free phase, no weight update occurs, and in the nudge phase, weights are updated through spikes based on a learning rule similar to Eq. (B6). They show that a version of STDP emerges from this learning rule.

4. The paper discusses the original two-phase CL update at length. However, multiple previous studies have proposed alternatives that are more compatible with physical systems. Two past proposals appear to be of particular relevance. First, the equilibrium propagation paper by one of the co-authors [30] suggests to integrate differences on the path from the free to the clamped state (see also Ref. [a]), which is conceptually not so far from what the authors propose here. Second, Baldi and Pineda study oscillatory teaching signals that make the system alternate between free and clamped states, as synapses continuously integrate phase-modulated synaptic currents. The relative merits of the new approach as well as its relationship to these older ideas should be discussed in depth.

We now highlight the differences between our work and Ref [a], which in our view is the most closely related algorithm in the literature. We refer to Ref [a]’s methods as Continual Equilibrium Propagation, or CEP for short. Nevertheless, there are strong differences between CEP and our current proposal, which are now highlighted in a new Related Work appendix:

d. Continual EP. Ernout *et al.* [50] rewrite the contrastive learning rule as

$$s_{ij}^{\text{free}} - s_{ij}^{\text{clamped}} = \int_{\text{free}}^{\text{clamped}} \frac{ds_{ij}}{dt} dt, \quad (\text{B4})$$

along any trajectory driving the system from free state to clamped state. Using this formulation of the learning rule, they propose an implementation of contrastive learning with continual weight updates in the clamped phase. The algorithm proceeds as follows:

1. **Free phase.** Let the network to reach the free state, while deactivating any weight update, i.e.

$$\frac{dw_{ij}}{dt} = 0. \quad (\text{B5})$$

2. **Clamped phase.** Starting from the free state, let the network reach the clamped state, while turning on the weight update

$$\frac{dw_{ij}}{dt} = \epsilon \frac{ds_{ij}}{dt} \quad (\text{B6})$$

along the trajectory from free state to clamped state.

In this approach again, one caveat is that it requires switching between two distinct weight update mechanisms. Our algorithm overcomes this hurdle.

We additionally clarify differences between the proposal of Baldi and Pineda and our present method:

e. Using oscillations. Baldi and Pineda [24] proposed to use a periodic nudging signal, e.g. a sinusoidal nudging signal with amplitude β and frequency w . The weights are then updated using the system's response to this periodic nudging signal, denoted as $s(t)$, as

$$\Delta w_{ij} = \epsilon \int_0^{2\pi/\omega} \sin(\omega t) s_{ij}(t) dt. \quad (\text{B7})$$

where $T = 2\pi/\omega$ is the period. Similar to other schemes, one caveat of this learning rule is that it requires being modulated by a time-varying factor $\sin(\omega t)$. [16] reused the idea of oscillations but, instead of the learning rule of Eq. (B7), they showed that in resistor, flow and elastic networks, the weight update for w_{ij} can be performed using the mean and amplitude of $s_{ij}(t)$. However, it remains unclear how the mean and amplitude of the signal can be easily extracted.

[a] Equilibrium Propagation with Continual Weight Updates. Maxence Ernout, Julie Grollier, Damien Querlioz, Yoshua Bengio, Benjamin Scellier, arXiv, 2020.

Reviewer #4 (Remarks to the Author):

The manuscript "Contrastive Learning through implicit non-equilibrium memory" proposes a novel way to implement a contractive divergence strategy to train quadratic energy functions without the need to store the "clamped" and the "free" configurations in memory. While the proposal may be interesting, I find the current version of the paper extremely difficult to read for non-specialists in the field of biological networks and difficult to apply to alternative problems in practice, so I cannot recommend its publication in Nature Communications.

Writing a manuscript for multiple distinct communities is a challenge. We thank the reviewer for raising key presentation issues that would enable clearer communication. We believe we have improved the manuscript now by following the reviewer's suggestions below and defining different terms in a more precise manner.

1. In particular, it is not clearly defined what the authors mean by "clamped" and "free" states, nor is the procedure by which they achieve these states clearly defined. In contrastive divergence, the positive and negative terms are usually averages of the correlations in the data set and in the equilibrium distribution of the model. I assume that the authors mean here the instantaneous configurations, i.e. a Hebbian learning procedure in combination with a dreaming strategy that is not explained. This might not be the case, because the way both types of states are reached is not explained, not even in the numerical test discussed.

We have now addressed this key presentation issue in two separate ways in the main text. First, we now clarify in the revised manuscript that the reviewer's guess is correct; we are focusing on deterministic versions of contrastive learning (CL), and hence free and clamped states are instantaneous configurations and not averages (line 108):

While CL was originally introduced and subsequently developed for stochastic systems (such as Boltzmann machines)[25, 32, 33], here we review the version of [22, 26] for deterministic systems (such as Hopfield networks). CL applies in systems described by an energy function E (more accurately, a Lyapunov function). The system may be supplied with a boundary input and evolves towards a minimum of E , called 'free state'. The system may also be supplied with a boundary desired output (in addition to the boundary input), driving the system towards a new energy minimum, called 'clamped state'.

Second, in the main text, we now discuss in much more detail the method by which those free and clamped states are generated (see edited text in "Performance on MNIST" section, line 283). We hope these changes make the paper easier to read for a wide readership.

2. I would like the authors to explain why it is much easier to switch between the two states and store the integral of the convolution with one kernel than to store two configurations. How do the neurons know how to switch back and forth between the two states? If the alternation is not complicated, why not simply store the difference between the two?

In our framework, the neurons themselves do not switch between Hebbian and anti-Hebbian modes as in prior work. Instead, an external ‘teacher’ applies time varying protocols to the ‘neurons’ (or molecular concentration or other physical degree of freedom), but the equation by which updates occur does not change through time (c.f. the methods outlined in response to R3.4).

A larger point of our paper is that there is no “storing” of the convolution needed - the convolution *is* the value of a dynamical variable in the system, when that variable is under the control of an integral feedback mechanism.

As we now emphasize in the rewritten explanation of our method (line 184):

This kernel convolution arises through the underlying physical dynamics in diverse physical and biological systems to be discussed later; in this approach, there is no explicit storage of the value of the signal $s_{ij}(t)$ at each point in time in specialized memory.

Thus the complexity demand on the system itself is greatly reduced, provided that integral feedback dynamics are available. We have additionally provided the equations for concrete physical implementations of integral feedback, and show how readout variables in these systems naturally compute convolutions of time-dependent forcings (see rewritten section “Realizing memory kernels through integral feedback”, and Eqs. 19-25).

The reviewer is right that all of the operations described by them (computing the convolution and then storing the integral thereof) would need to be implemented if our algorithm were run on a digital computer. However, our objective here is to enlarge the universe of “analog hardware” that can learn despite not having a GPU unit, not to propose another algorithm for digital hardware.

As we now clarify in our revised introduction (line 73):

Here, our primary contribution is to show how a ubiquitous process - integral feedback control - can allow for contrastive learning without the complexities of explicit memory of free and clamped states or switching between Hebbian and anti-Hebbian update modes. Further, in our method, computing the contrastive weight update signal and then performing the update of weights are not separate steps involving different kinds of hardware but are the same unified in situ operation.

3. The numerical test using this training scheme to predict the labels from MNIST is incomprehensible to me. How do you go from a pairwise model to predicting 10 labels? The authors should explain the set-up and how this machine was trained. Can their networks be trained to work as generative models or at least store patterns?

We thank the reviewer for encouraging clarification of our MNIST training example, which shows how a neural network trained with our proposed method can store and recall patterns deterministically (e.g. as Hopfield spin-glass networks).

We have now extensively reworked our MNIST example section to be explicit in what was done. For example, as we now say in regards to predicting labels:

PERFORMANCE ON MNIST

By coupling together a network of synapses with non-equilibrium memory kernels, we were able to train a neural network capable of classifying MNIST. In particular, we adapted the Equilibrium Propagation (EP) algorithm [26], a contrastive learning-based method that ‘nudges’ the system’s state towards the desired state, rather than clamping it as is done in standard contrastive Hebbian learning (CHL) [22]. We used EP instead of CHL due to its better properties and its superior performance in practice [58, 59]. EP makes weight updates of the form:

$$\Delta w_{ij} = \epsilon(s_{ij}^{\text{nudge}} - s_{ij}^{\text{free}}). \quad (8)$$

Normally the EP method requires storing the states s_{ij}^{nudge} and s_{ij}^{free} in memory, computing the difference and then updating weights w_{ij} ; our proposed TCL method will accomplish the above EP weight update, without explicitly storing and retrieving those states.

In order to process MNIST digits, we utilize a network architecture with three types of symmetrically-coupled nodes. Each node carries internal state x and activation $\eta = \text{clip}(x, 0, 1)$. Nodes belong either to a 784-node input layer (indexed by i), a 500-node hidden layer (indexed by h), or a 10-node output layer (indexed by o). Nodes are

constant values over time, $x_i(t) = v_i^{\text{image}}$. For inference, we allow the hidden- and output-layer activations to adjust in response to the fixed input nodes, minimizing Eq. 9. Once a steady-state is reached, the network prediction is given by looking at the states of the 10 output nodes, $\{x_o\}_{o=0,\dots,9}$, and interpreting the index of the maximally activated output node as the input image label.

For the full edit, including how our neural network was trained, please see red text in “Performance on MNIST” section, line 282.

4. Can this memory method be related to the recently proposed out-of-equilibrium recipe for training energy-based models in general? (“Explaining the effects of non-convergent sampling in the training of Energy-Based Models” ICML 2023)

We focus more on deterministic inference with energy-based models, as we now emphasize in a new section “Background: contrastive learning”; the ICML 2023 work is cited there in reference to approaches to stochastic inference in a contrastive framework (line 108):

While CL was originally introduced and subsequently developed for stochastic systems (such as Boltzmann machines)[25, 32, 33], here we review the version of [22, 26] for deterministic systems (such as Hopfield networks).

Additionally, our rewritten introduction clarifies that one of the main contributions of our paper is to propose a physical process by which the many learning rules involving the contrastive difference of two terms (e.g. Eq. (2) in the referenced ICML paper) can be implemented in extremely low-level, simple physical systems. It is not clear to us how the scheme outlined in the reference, involving sampling over multiple MCMC runs and computing corrections before making a weight update, could be naturally realized in such a context.

Reviewer #5 (Remarks to the Author):

Dear Editor and Authors,

This is an interesting paper. However, in my view, there is significant margin for improvement. Please find my feedback below.

1. Most importantly, the proposed approach has very strong links to STDP, but the authors only mention STDP once and very briefly, only to assert that the manuscript's approach is significantly different. This would need a rigorous substantiation. I must say that I doubt that an attempt to do this will result in a truly significant difference. Specifically, it appears that the mechanism described in this paper is a rate-based description of STDP's outcome, if the authors' s_{ij} is considered as a rate. There are multiple prior works that show the relationship between spike-based and rate-based learning. These works include rate-based interpretations of STDP (e.g. [r1]), and also the reverse, i.e. derivations of optimal STDP kernels from rate-encoded inputs (e.g. Section S6.1 in [r2]). This latter type of prior work also relates to the section of the present manuscript that examines the kernel shape or area. That section also relates to a further large body of literature that examines the effect of STDP kernel shapes on learning but is uncited.

We have now clarified more concretely the relationship between STDP and our current proposed method. In particular, the formulation of Xie and Seung (Neurips, 1999) provides a natural comparison point between both spike-/rate-based descriptions of STDP and our method, being written in terms of kernel convolutions, but also reveals their fundamental differences. We have highlighted this both in the main text:

Additionally, our rule superficially resembles spike-timing dependent plasticity (STDP)[41, 51, 52], in particular how our rule has a representation involving a non-monotonic kernel K . While STDP-inspired rule[53-55] and other rules involving competing Hebbian updates with inhibitory neurons[56,57] have shown great promise as local learning paradigms, they are adapted to settings where synapses are asymmetric and can distinguish differentials between the timings of pre- and post-synaptic neuronal activations. Our rule only involves signals at the same moment in time t and can only result in symmetric interactions w_{ij} . We are not aware of any direct relationship between work on STDP rules and the proposal here; see Appendix B for details.

And in the new related works appendix:

f. Connection with STDP. Spike-timing-dependent plasticity (STDP) (Bi and Poo [51]) can be formulated as a learning rule for asymmetric networks where a synapse distinguishes between the time of presynaptic spikes t_k and the time of postsynaptic spikes t_j :

$$\Delta w_{ij} = f(t_j - t_k) \quad (\text{B8})$$

where f has a positive lobe for positive arguments and a negative lobe for negative arguments, and approximately vanishes outside a pairing range $[-\tau, \tau]$.

A large body of work has interrogated the equivalence of spike- and rate-based formulations of STDP [54, 55]. In particular, Xie and Seung [42] considered additive STDP updates for δ -function timeseries $s_j(t)$, written as:

$$\Delta w_{ij} = \int_0^T dt_k \int_{-\infty}^{\infty} dt_j f(t_j - t_k) s_j(t_j) s_k(t_k) \quad (\text{B9})$$

over a time period $[0, T]$. They showed that Eq. B9 can be approximated as rate-based formulations depending on firing rates $\nu_j(t)$ when rates of spiking vary slowly compared to the pairing range:

$$\Delta w_{ij} = \int_0^T dt [\beta_0 \nu_j(t) + \beta_1 \dot{\nu}_j(t)] \nu_k(t) \quad (\text{B10})$$

where β_0 is the integral of f over the pairing range, and β_1 is the first moment of f over the pairing range.

In the case where $\beta_0 = 0$ and $\beta_1 > 0$, Eq. B10 can be interpreted as the asymmetric version of the nudge phase update of Continual EP (Eq. B6):

$$\frac{dw_{ij}}{dt} = \beta_1 \dot{\nu}_j(t) \nu_k(t). \quad (\text{B11})$$

Martin *et al.* [53] proposed a spiking version of contrastive learning to train spiking networks with bidirectional weights. Similar to the Continual EP approach, in the free phase, no weight update occurs, and in the nudge phase, weights are updated through spikes based on a learning rule similar to Eq. (B6). They show that a version of STDP emerges from this learning rule.

Further aspects of the manuscript that I recommend improving are the following:

2. Equilibrium Propagation is used and modified, but it and its specific modifications are not described.

We have now substantially increased the level of detail in our main text description of our EP MNIST experiment, see also responses to Reviewer #4:

PERFORMANCE ON MNIST

By coupling together a network of synapses with non-equilibrium memory kernels, we were able to train a neural network capable of classifying MNIST. In particular, we adapted the Equilibrium Propagation (EP) algorithm [26], a contrastive learning-based method that ‘nudges’ the system’s state towards the desired state, rather than clamping it as is done in standard contrastive Hebbian learning (CHL) [22]. We used EP instead of CHL due to its better properties and its superior performance in practice [58, 59]. EP makes weight updates of the form:

$$\Delta w_{ij} = \epsilon(s_{ij}^{\text{nudge}} - s_{ij}^{\text{free}}). \quad (8)$$

Normally the EP method requires storing the states s_{ij}^{nudge} and s_{ij}^{free} in memory, computing the difference and then updating weights w_{ij} ; our proposed TCL method will accomplish the above EP weight update, without explicitly storing and retrieving those states.

In order to process MNIST digits, we utilize a network architecture with three types of symmetrically-coupled nodes. Each node carries internal state x and activation $\eta = \text{clip}(x, 0, 1)$. Nodes belong either to a 784-node input layer (indexed by i), a 500-node hidden layer (indexed by h), or a 10-node output layer (indexed by o). Nodes are

connected by synapses only between adjacent layers (i and h , h and o), with no skip- or lateral-layer couplings. The neural network dynamics minimize the energy:

$$E(x) = \frac{1}{2} \sum_n x_n^2 - \frac{1}{2} \sum_{n,m} w_{nm} \eta_n \eta_m - \sum_n b_n \eta_n, \quad (9)$$

where b_n is the bias of node n , w_{nm} is the weight of the synapse connecting nodes n and m , and the indices n, m run over the node indices of all layers i, h, o .

We represent each 784-pixel grayscale MNIST image as a vector $v^{\text{image}} = \{v_i^{\text{image}}\}_{i=0, \dots, 783}$. For each MNIST digit v^{image} , we hold the states of the 784 input nodes at

constant values over time, $x_i(t) = v_i^{\text{image}}$. For inference, we allow the hidden- and output-layer activations to adjust in response to the fixed input nodes, minimizing Eq. 9. Once a steady-state is reached, the network prediction is given by looking at the states of the 10 output nodes, $\{x_o\}_{o=0,\dots,9}$, and interpreting the index of the maximally activated output node as the input image label.

During inference, the network minimizes an energy function which does not vary in time, subject to the constraint that $x_i(t) = v_i^{\text{image}}$. During training, the same constraint $x_i(t) = v_i^{\text{image}}$ applies, but our network instead is subjected to a time-varying energy function:

$$F(x;t) = E(x) + \frac{\beta(t)}{2} \sum_{o=0}^9 (x_o - v_o^{\text{label}})^2. \quad (10)$$

Here, v^{label} is the one-hot encoding vector for the corresponding label of the MNIST digit. The time-dependent training protocol $\beta(t)$ is the asymmetric sawtooth function which smoothly interpolates between 0 and a maximal value $\beta_{\text{max}} < 1$. One portion of the sawtooth is characterized by the fast timescale τ_f , and the other portion by the slow timescale τ_s . We assume that both these timescales are quasi-static compared to any system-internal energy relaxation timescales[60]. This restriction to adiabatic protocols is a potential limitation to contrastive methods in general; details depend on the system and some works present viable workarounds[25, 61].

Each time the training protocol $\beta(t)$ completes a cycle, the network weights are updated according to Eq. 6, with $\eta_n \eta_m$ as the necessary synaptic current s_{nm} . After completing multiple sawtooth cycles, we switch the inputs x_i to a new MNIST image and repeat the training process of manipulating the x_o through the time-varying energy function $F(x;t)$. See Appendix C for further detail, including specifications of K and g for Eq. 6.

We find that, after training for 35 epochs, our classification error drops to 0, and we achieve an accuracy of 95% on our holdout test dataset (Fig. 3B). Our results demonstrate the feasibility of performing contrastive learning in neural networks without requiring explicit memory storage, but leave open the question of limitations on our approach. We consider performance limitations at the level of a single synapse in the following sections.

3. The aspects of the manuscript that focus on "dissipation" seem interesting, but dissipation as a concept is not introduced. For example, what is dissipated? What is meant by "dissipative integral feedback dynamics"?

We use dissipation to mean energy dissipation; we now explicitly refer to dissipation as “energy dissipation” in several parts of the paper. As we explain further in the paper, this energy dissipation is the amount of energy that will have to be used up to implement this algorithm due to a fundamental Landauer-like energy cost of computation; thus it is the familiar energy cost of computation (‘dissipation’ because when energy - e.g., chemical energy in a battery - is used up to do a computation, the result is heat). E.g. line 411:

In other words, to perform increasingly accurate inference, systems need to consume more energy (e.g. electrical energy in neuromorphic systems, ATP or other chemical fuel in molecular systems), which is then dissipated as heat.

We finally clarify that this dissipation can be rigorously computed in the master equation context we study (line 487):

In the Markov chain context, we can rigorously compute energy dissipation $\sigma = \sum_{i>j} (r_{ij}p_j - r_{ji}p_i) \ln\left(\frac{r_{ij}p_j}{r_{ji}p_i}\right)$.

4. The manuscript mentions that Hebbian and anti-Hebbian plasticity are commonly applied in separate phases, but that does not acknowledge previous work that has indeed applied them in a single phase and showed that this works relatively well (e.g. [r3, r4]). The present work seems to also use a single learning mode at the synapse level, but it does switch modes by varying the clamping signal, and by expanding an individual training example from a single timestep to an extended duration. This is not commented on. What is the advantage compared to these other works?

We now comment on the relationship of these learning paradigms to ours in the new related works appendix:

g. Competitive Hebbian updates. In a directional context with presynaptic neuron i and postsynaptic neuron j , Journé *et al.* [57] consider weight updates of the form:

$$\Delta w_{ij} = \epsilon y_j (x_i - u_j w_{ij}) \quad (\text{B12})$$

where x_i is the activation of neuron i and u_j is the weighted input to neuron j ; y_j is a winner-take-all factor across all K neurons in a given layer, parameterized by a factor b :

$$y_j = \frac{b^{u_k}}{\sum_{l=1}^K b^{u_l}}. \quad (\text{B13})$$

This method of learning is well-suited to a neuronal context. However, implementing it in less internally complex physical and biological systems would be a non-trivial task, requiring e.g. the engineering of equivalents to pre- and post-synaptic neurons, as well as the computation of the y_j Boltzmann factor. The latter could be accomplished potentially through lateral inhibition within layers[56], at the cost of making architectural assumptions that would narrow the range of natural settings the method could be implemented in.

5. Even compared to Equilibrium Propagation, and the other algorithms that do involve two phases, could the authors increase the clarity on the benefit of the present mechanism? The key seems to be in the contrast between "implicit" and "explicit" memory, in the authors' terms. Can this be clarified further?

We thank the reviewer for encouraging us to be more clear with the benefits of our proposed mechanism. As we have now clarified, by an "implicit" memory mechanism, we mean one which:

1. Does not need to switch synapses between Hebbian and anti-Hebbian learning rule phases (which requires a global signal);
2. Does not need separate memory units to store information about each of the two phases;
3. Does not require separate calculation of update value and actual updating; the calculation and updating happens at once.

We now clarify this in our Introduction (line 60):

Therefore, a central obstacle for autonomous physical systems to exploit contrastive learning is that weight updates require a comparison between free and clamped states,

but these states occur at different moments in time. Such a comparison requires memory at each synapse to store free and clamped state information in addition to global signals that switch between these free and clamped memory units and then retrieve information from them to perform weight updates[27, 28]. These requirements make it difficult to see how contrastive learning can arise in natural physical and biological systems and demand additional complexity in engineered neuromorphic platforms.

Please also see response 5.8.

6. Experiments are very limited. Only a one-hidden-layer network is used, only MNIST classification is tested, and there are no comparisons with alternative learning rules.

1. As suggested by the reviewer here and also in comments 4.3, 4.4, and 5.1, we have now added a Related works section that makes comparisons of the pros/cons of our approach with other similar learning rules (see “Related work summary” line 254 and “Related work” Appendix B).
2. In terms of quantitative comparisons, as our revised manuscript points out, the time cost is indeed such a comparison to EqProp implemented with explicit memory instead.
 - a. For example in the Discussion (line 734): “Our framework has several limitations quantified by speed-accuracy and speed-energy tradeoffs derived here, in addition to the general concerns about adiabatic protocols in contrastive methods[25, 60]. The time costs in our scheme (e.g. those shown in Fig. 4) are inherently larger compared to other Equilibrium Propagation-like methods which use explicit memory and global signals to switch between wake and sleep, thereby avoiding slow ramping protocols.”
3. Note that our goal is not to produce an algorithm for digital hardware but for highly constrained analog hardware, such as specialized neuromorphic hardware and in natural systems (E.g., the flow network of *Physarum polycephalum* learning a new stimuli-response behavior). While it might be natural to test algorithms for digital hardware on other standard benchmark datasets, they are not as relevant to some of the natural systems we detail in our “Realizing memory kernels through integral feedback” section. Instead, we have now expanded the quantitative models of how the paradigm proposed here would work in these specific physical and biological systems (see response to 1.1, as well as expanded “Realizing memory kernels through integral feedback” section).

7. The synapses are bidirectional, which is not plausible biologically.

While biological neurons are indeed not bidirectional, we now have an extended section detailing real physical and non-neural biological systems described by energy functions that are covered by the scope of this work (“Realizing memory kernels through integral feedback” line 524). Thus we do not consider it a weakness that our synapses are bidirectional; many of the

low-complexity physical systems which we would be interested in, e.g. elastic materials, chemical reaction networks, flow networks, etc., are in fact symmetric; some of these systems are biologically relevant, just not in a neuronal context.

In the main text of the paper, we have written a short section which points readers to a longer appendix where we compare our approach to those already in the literature (see point 5.1, "Related Work Summary" line 254). This short pointer highlights the differences between our approach and those involving directional neurons.

8. More generally, it would be very helpful if the authors' response could also include a statement on the broader significance and impact of this manuscript.

We have streamlined the Introduction and emphasized there what we consider to be our primary contribution to the literature (line 73):

Here, our primary contribution is to show how a ubiquitous process - integral feedback control - can allow for contrastive learning without the complexities of explicit memory of free and clamped states or switching between Hebbian and anti-Hebbian update modes. Further, in our method, computing the contrastive weight update signal and then performing the update of weights are not separate steps involving different kinds of hardware but are the same unified *in situ* operation. This approach to contrastive learning, which we call 'Temporal Contrastive Learning through feedback control', allows a wide range of physical and biological platforms without central processors to 'physically learn' (i.e., autonomously learn) novel functions through contrastive learning methods.

*Sincerely,
Timos Moraitis*

References

*[r1] Burkitt, Anthony N., Hamish Meffin, and David B. Grayden. "Spike-timing-dependent plasticity: the relationship to rate-based learning for models with weight dynamics determined by a stable fixed point." *Neural Computation* 16.5 (2004): 885-940.*

*[r2] Moraitis, Timoleon, Abu Sebastian, and Evangelos Eleftheriou. "Optimality of short-term synaptic plasticity in modelling certain dynamic environments." *arXiv preprint arXiv:2009.06808* (2020).*

*[r3] Krotov, Dmitry, and John J. Hopfield. "Unsupervised learning by competing hidden units." *Proceedings of the National Academy of Sciences* 116.16 (2019): 7723-7731.*

[r4] Journé, Adrien, Hector Garcia Rodriguez, Qinghai Guo, and Timoleon Moraitis, "Hebbian Deep Learning Without Feedback." *The Eleventh International Conference on Learning Representations (ICLR 2023)*.

Reviewer #5 (Remarks on code availability):

I could not find any code attached.

We apologize for the oversight; code can now be found at:

1. <https://github.com/falkma/ContrastiveMemory-Exp>
2. <https://github.com/atstrupp/ContrastiveMemory-SynapseAnalysis>

Reviewer #1 (Remarks to the Author):

The authors have clarified some of the issues I raised in the first round of reviews. While their responses made the scope of the manuscript clear, they also raise major questions that need to be resolved.

Thank you for taking the time to review our work and suggest improvements. We are glad many of the issues regarding the scope and motivation of this work have been clarified.

Most importantly, the authors claim that they are not developing an algorithm that can compete with the dominant backpropagation (that is usually run on digital hardware), but they are focused on optimizing an algorithm for non-digital (mostly analog hardware). To that end, the authors need to provide clear and fair comparisons among the main existing non-backpropagation algorithms. Please compare the performance, time/cycles taken, energy consumed (using simple assumptions on analog hardware), etc. You could use a small data set to demonstrate your performance metrics. This exercise need not be very laborious - it can be on toy-sized models and datasets. But this exercise is important in a paper published in a general-audience journal, without which it becomes a mere mathematical demonstration.

We are glad it was clarified that the goal isn't an algorithm for digital hardware but rather how non-backprop learning can be exploited by analog systems that range from natural molecular networks in cells to engineered neuromorphic hardware.

We have now addressed the request for comparisons with an extended new section in the paper. We would like to clarify that there has not really been any concrete proposal of how to implement these algorithms in entirely analog hardware. Some aspect or the other involves a digital operation - e.g., typically storing sleep and wake states in digital memory, subtracting them digitally and then making weight updates. As a consequence, while we have made the needed comparisons, these comparisons have a bit of an apples-to-oranges flavor since our proposal is the first truly analog implementation of EP and related contrastive methods while the comparison points have a digital element.

Nevertheless, we agree with the reviewer that a discussion of such comparisons is important. In the edited paper, we make the following comparisons:

1. Our method to prior proposals for EP based on local digital memory
2. Our method to prior proposals for EP based on two copies of the system
3. Our method to prior proposals for contrastive training on a digital computer.

Hence we have now explicitly quantified the time and energy costs of our proposed fully analog implementation of contrastive learning, so it is easy to compare these results to other algorithms when analog implementations for them come online. We report these results in a new Appendix C, "Quantitative time and energy comparison of TCL vs other contrastive methods":

1. For time, we find that our method scales mainly with the kernel timescale τ_K , whereas other contrastive methods scale as the linear combination of the system relaxation timescale τ_{relax} and the timescale of fetching memory, τ_{fetch} .
2. For energy per cycle, our method scales with the number of synapses $N_{synapse}$ and the total cycle time ($\tau_f + \tau_s$). In contrast, explicitly contrastive methods dissipate energy in writing and erasing memory, which has a Landauer lower bound which scales with the number of N_{nodes} and n , the number of bits to which the memory of the states are stored.

For TCL:

1458 For TCL, the time cost of computing Eq. 6 is set
 1459 by the dynamic range to which we want to be able to
 1460 measure the contrastive difference. For example, if we
 1461 want to be able to distinguish differences between free
 1462 and clamped states by a ratio of 10^{-2} , this would re-
 1463 quire $\sim 10^3 \tau_K$, where τ_K is the timescale of the physical
 1464 processes generating integral feedback in the analog sys-
 1465 tem (Fig. 8). Similarly, there is a cost associated with
 1466 the energy dissipation rate σ of running integral feed-
 1467 back in each synapse, which takes about $\sim 5 - 10kT$
 1468 in order to achieve 10^{-2} relative error to the explicit
 1469 contrastive update Eq. 6 (Fig. 6). The total energy
 1470 cost of running the network per cycle therefore would be
 1471 $\sim 10kTN_{synapse}(\tau_f + \tau_s)$, with the number of synapses
 1472 $N_{synapse}$ scaling with the number of nodes N or N^2 de-
 1473 pending on network connectivity.

For other contrastive algorithms:

1474 For explicitly contrastive implementations, one time
 1475 cost of computing Eq. 6 is instead the time of storing
 1476 each state and retrieving it to perform the difference op-
 1477 eration. For memory stored on a digital device locally at
 1478 each node, this is a constant time, τ_{fetch} . For fully digital
 1479 implementations (e.g. simulations of EP methods on a
 1480 computer) this time is extensive in N , i.e. $N\tau_{fetch}$. This
 1481 is in addition to the time cost of relaxation of the physi-
 1482 cal state of the system in response to alternating between
 1483 clamping and releasing the output nodes, which we label
 1484 τ_{relax} . Therefore, to simulate, for example Equilibrium
 1485 Propagation[26] on a computer would naively take a time
 1486 $2\tau_{relax} + N\tau_{fetch}$ per cycle. In a twinned circuit device as
 1487 constructed in Refs. [47, 108], the time per cycle would
 1488 be $\tau_{relax} + \tau_{fetch}$.

Finally, we emphasize that the other proposed implementations of contrastive learning suffer from important practical problems in terms of implementation. Hence our work's key strength is that it is not a "mere mathematical demonstration" but considers the physical constraints of diverse systems. See SI Appendix C for new analyses.